# Malignant Superficial Mesenchymal Tumors in Children

**DOI:** 10.3390/cancers14092160

**Published:** 2022-04-26

**Authors:** Philippe Drabent, Sylvie Fraitag

**Affiliations:** 1Department of Pathology, Necker-Enfants Malades Hospital, APHP, 75015 Paris, France; philippe.drabent@aphp.fr; 2Faculté de Médecine, Université de Paris, 75005 Paris, France

**Keywords:** children, skin, sarcoma, mesenchymal tumors, diagnosis, histology, genetics

## Abstract

**Simple Summary:**

Malignant tumors of the skin and subcutaneous tissue are rare in children. Most of these cancers are mesenchymal tumors and among these tumors, most have an intermediate malignant potential or are considered low grade sarcomas. In addition, some sarcomas of deep soft tissues may also involve the skin by contiguity. This review aims to sort out the diversity of these malignant mesenchymal tumors in children, with a particular focus on clinical features that may be useful for clinicians (especially age at presentation) and on the newest entities and genetic data.

**Abstract:**

Malignant superficial mesenchymal tumors are a very diverse group of neoplasms with few clinical and radiological discriminatory factors. Hence, some of these cancers are rarely suspected based on clinical and radiological grounds, others may be easily misdiagnosed, and the histological analysis of a biopsy or resection is central in the diagnostic process. In children, the age at presentation is a major element of the differential diagnosis. Some tumors have a very distinct epidemiology, while others may be seen at any age. More recently, the advances in molecular biology have greatly improved the diagnosis of mesenchymal tumors and new entities are still being described. In the present review, we provide an overview of the diversity of malignant superficial mesenchymal tumors in children, including new and/or rare entities. We discuss the important diagnostic features, be they clinical, histological, or molecular. Special attention was given to the genetic features of these tumors, particularly when they were helpful for the diagnosis or treatment.

## 1. Introduction

This review focuses on primitive malignant mesenchymal tumors of the skin in children. Neuroectodermal tumors are included here as well, since they share some characteristics with soft tissue tumors of neural differentiation. Superficial leiomyosarcoma and epithelioid hemangioendothelioma are far too exceptional in children to have been included hereafter. Metastases are not discussed, or only briefly for differential diagnosis. Pseudotumors and benign tumors do not fall into the topic of this review.

There are few studies that have focused on the epidemiology of malignant mesenchymal tumors of the skin specifically in children. It is important to bear in mind that, in children and adolescents, soft tissue tumors of intermediate malignant potential are by far more frequent than high grade sarcomas. In one American study, the most prevalent cancer was rhabdomyosarcoma (RMS), however, it was never primitively cutaneous and rather arose from the subcutis or deep soft tissues and involved the skin secondarily [1]. Although primitively cutaneous RMS is exceedingly rare, the fact that deeper seated RMS may involve the skin prompted us to discuss this cancer quite extensively. As expected, the second most prevalent cancer in the same study was dermatofibrosarcoma protuberans. However, this was followed by synovial sarcoma, primitive neuroectodermal tumor, and malignant peripheral nerve sheath tumor (MPNST), all tumors that, in our experience, are very rare in the skin and arise more deeply. This underlines how there may be wide differences between centers, and how soft tissue sarcomas remain an important differential diagnosis. Even more significant for clinical practice, the age range is of the utmost importance when discussing mesenchymal tumors of the skin in children. This was highlighted by Liszewski et al., and also by the French network of cancer registries (FRANCIM) [2]. Soft tissue tumors that are predominate in the neonatal period (infantile fibrosarcoma, kaposiform hemangioendothelioma, melanotic neuroectodermal tumor of infancy, and others) are very different from those that are seen mainly in adolescents. Cutaneous soft tissue tumors of adolescents are close to that of adults, with the particularity of some genetic predispositions or medical contexts that may explain the early presentation of the cancer (e.g., MPNST in children with neurofibromatosis type 1 or with a history of radiation therapy). For practical purposes, the tumors discussed hereafter are classified according to their line of differentiation. In addition, the age ranges of these tumors, which may be useful in clinical practice, are outlined in Figure 1. Similarly, the different genetic anomalies that are currently recognized in these tumors are summarized in Table 1.

## 2. Fibroblastic and Fibrohistiocytic Tumors

### 2.1. Angiomatoid Fibrous Histiocytoma

Angiomatoid fibrous histiocytoma (AFH) can arise at any age in children and young adults. It affects either sex equally and is located in the limbs or trunk, exceptionally in the head and neck. AFH arises in the subcutaneous tissues, less commonly in deeper soft tissues, and very rarely in the dermis alone. Unusual visceral or deep locations have also been reported: lung, brain, mediastinum, retroperitoneum, ovary, or bone. The tumor presents as a slowly growing, usually non-tender nodule less than 2 cm in diameter, although some tenderness or pain may be present. Systemic symptoms have been described in some cases, for example, weight loss, fever, or anemia, but were not found in all series [3]. Imaging characteristics may be misleading. AFH is isointense to muscle on T1-weighted images, heterogeneously hyperintense on T2-weighted images, and shows variegated enhancement after gadolinium injection. It also often features a pseudo-capsule, cystic areas (sometimes with fluid-fluid levels), and peritumoral edema. The most frequent suggested diagnoses on MRI are those of hemangioma and vascular malformation [4,5].

Microscopically, AFH is well circumscribed, often with a partial fibrous pseudo-capsule. It is made of multiple nodules or lobules with characteristic blood-filled pseudo-angiomatous spaces, often more prominent at the periphery of the tumor (Figure 2A,B). Up to one third of cases may have completely solid morphology without pseudo-angiomatous spaces. A dense lymphocytic infiltrate is also frequent at the periphery of the lesion, often with lymphoid follicular hyperplasia and germinal center formation (Figure 2C). In the nodules, the tumor cells have a fibroblastic-like or histiocytoid morphology (Figure 2D). Mitoses are sparse but atypical forms may be found, with no clinical implication. No histopathologic feature can predict behavior. Immunophenotype is not specific but supportive findings are positivity for desmin or rarely other myoid differentiation markers (SMA, calponin); positivity for EMA; positivity for CD68; and cytoplasmic positivity for CD99 (Figure 2E,F). All vascular markers, cytokeratins, and S100 are negative [3,4].

The most common chromosomal translocation in AFH is t(2;22)(q33;q12) resulting in the *EWSR1-CREB1* fusion gene. Other translocations are t(12;22)(q13;q12) (*EWSR1-ATF1*) and t(12;16)(q13;p11) (*FUS-ATF1*). CREB1 and ATF1 are both members of the cAMP response element binding protein (CREB) family. *EWSR1-CREB1* and *EWSR1-ATF1* fusions are also seen in clear cell sarcoma and other non-cutaneous tumors [4].

Most AFH have a benign clinical course, with local recurrence in up to 15% of cases. Metastases to local lymph nodes occur in less than 5% of cases. Distant metastases to the lungs, liver, or brain are exceptional, with only very rare deaths occurring decades after initial presentation. Therefore, the recommended management of AFH is complete local excision, with adjuvant radiation therapy or chemotherapy being reserved only to metastatic or unresectable cases [4,5].

### 2.2. Infantile Fibrosarcoma

Infantile fibrosarcoma is the most frequent malignant soft tissue tumor of the skin and subcutis before the age of 1 [1]. Around a third of cases are congenital [6]. The median age is 3 to 4 months and 75% of cases occur in the first year of life [7].

About 90% of dermatologic cases are located in the subcutaneous tissue, the remaining cases resulting from the involvement of the subcutis by a more deeply located tumor [1]. There is a slight predilection for the lower extremities but all locations are seen [1,6]. The tumor is typically a painless, rapidly growing nodule or mass, ranging from 1 to >15 cm [8,9]. There is no specific imaging finding [9]. Ultrasound (US) typically shows a well-demarcated, homogeneous mass (54%), but heterogeneity is quite frequent (46%), consisting of hemorrhage foci or more rarely cystic components. On MRI, most tumors have isointense T1 and hyperintense T2 signal intensity with heterogeneous enhancement [9]. When dilated vessels and hemorrhage are prominent, the tumor may resemble a vascular tumor (Figure 3A). Cases with mild thrombocytopenia have been described, which should not be confused with Kasabach–Merritt phenomenon [8,10].

The diagnosis rests on the pathological analysis of a biopsy (including fine-needle biopsy), which should be as deep as possible. Indeed, a superficial sampling of the tumor may show misleading aspects, such as lipofibromatosis-like [11], hemangioma-like features [10,12] or fat necrosis. This can be avoided by favoring a deep surgical biopsy or fine needle biopsy (Figure 3B,C). More often than not, infantile fibrosarcoma demonstrates a cellular, monomorphic neoplasm made up of ovoid to spindle cells arranged in compact sheets or long fascicles, sometimes assuming the typical herringbone pattern (Figure 3D). Mitoses can be sparse or numerous from one tumor to another and in the same tumor, with no prognostic impact. Necrosis is possible and has no prognostic implication either. The tumor vessels may be numerous. In such cases, the tumor might even be misdiagnosed as a vascular anomaly. The immunophenotype is not specific. The tumor may show variable and focal expression of SMA, S100, CD34, and rarely desmin (or none) [13]. Expression of pan-TRK (neurotrophic tyrosine receptor kinase) is helpful for screening tumors with a *NTRK* gene rearrangement [14]. Indeed, most infantile fibrosarcomas exhibit an *ETV6-NTRK3* gene fusion resulting from the t(12;15)(p13;q25) chromosomal translocation. Other *NTRK3* fusion partners include *EML4* [15], *RBPMS* [16], or *SPECC1L* [17]. Other tyrosine kinase genes are involved in fusions, including *NTRK1*, *NTRK2*, *RET*, *MET*, and *RAF1*, with a variety of partners [16,18,19,20]. *BRAF* fusions, complex deletions, and point mutations have also been evidenced, inducing constitutive activation of the RAF-MEK-ERK pathway [18,19]. Interestingly, other spindle cell tumors harboring *RAF1*, *BRAF*, and *NTRK1/2* fusions and further characterized by S100 and CD34 co-expression (discussed hereafter, see Section 2.3), demonstrate a close proximity to some infantile fibrosarcomas by RNA unsupervised hierarchical clustering analysis [21]. This suggests a common pathogenesis between these tumors, or even a common spectrum of low-grade spindle cell tumors with activation of the RAF-MEK-ERK pathway.

Infantile fibrosarcoma has an overall favorable prognosis. It is locally aggressive and very rarely metastasizes. Metastases to the central nervous system, the lungs, and lymph nodes have been reported a few times [6,22,23,24,25,26,27]. There are rare reports of metastases to the heart, bones, liver, and adrenal glands [24,28,29]. The standard treatment includes complete surgical excision (as often as possible), associated with chemotherapy when necessary (incomplete or impossible excision). Targeted tyrosine kinase inhibitor therapies (larotrectinib, entrectinib, selitrectinib, and repotrectinib) seem to be promising alternatives [27,30,31,32].

From a histological point of view, the differential diagnosis includes *NTRK*-rearranged spindle cell neoplasms (see below), dermatofibrosarcoma protuberans (DFSP, see Section 2.6), desmoid fibromatosis, synovial sarcoma (see Section 7.3), the exceptional *EWSR1-SMAD3* positive fibroblastic tumor (see Section 2.4) and *ALK*-rearranged infantile fibrosarcoma-like tumor (see Section 2.5).

### 2.3. NTRK-Rearranged Spindle Cell Neoplasms

This group of tumors is emerging as an entity based on molecular grounds. More than 55% of pediatric cases occur in the first year of life [16,20]. Some cases are congenital [33]. However, all ages can be affected [21,33,34].

Most *NTRK*-rearranged spindle cell neoplasms occur in the superficial soft tissues of the extremities, other locations being the trunk and scalp. They present as a subcutaneous nodule or mass ranging from 1 to 6 cm in greatest dimension [35]. Imaging characteristics of these tumors have never been studied; however, it is likely that they overlap with infantile fibrosarcoma, lipofibromatosis, and malignant peripheral nerve sheath tumor (MPNST), and thus, are not specific.

Indeed, *NTRK*-rearranged spindle cell neoplasms include a wide variety of morphologies with tumors described as infantile fibrosarcoma-like, inflammatory myofibroblastic tumor-like (IMT-like), lipofibromatosis-like neural tumor (LPF-NT), and MPNST-like [36]. Microscopically, the most consistent features are an infiltrative growth pattern within subcutaneous fat, dense spindle cells haphazardly arranged or in fascicles, elongated nuclei with mild atypia and hyperchromasia, inconspicuous nucleoli, and low mitotic activity (Figure 4A,B) [16,21,33,34,35,36]. A prominent inflammatory infiltrate has been described in IMT-like cases [16,21,34,37]. The clue to the diagnosis is the co-expression of CD34 and S100 by immunohistochemistry. The expression may be focal and vary among cases [34,35]. The retained expression of H3K27me3 can help in the differential diagnosis with a MPNST, a differential that may be considered only in adolescents. Screening with an anti-NTRK1 or panTRK antibody is also helpful (Figure 4C). The definitive diagnosis is made by molecular testing for *NTRK1* gene fusions (in most cases, Figure 4D). The fusion partners include *LMNA*, *TPM3*, *TPR*, and *SQSTM1* [16,34]. Other gene rearrangements have been identified in children, involving *NTRK3* with *EML4* or *KHDRBS1* as partners [21,38,39]. We have already discussed the possible relation between *NTRK*-rearranged spindle cell neoplasms and infantile fibrosarcoma (see Section 2.2).

In children, for those tumors located in superficial soft tissues, the prognosis is favorable in most instances. The clinical course is marked by local recurrence, sometimes repeated [16,21]. Metastases are rare but have been recorded in one series in 3 children (out of 22) aged 0 to 5 years. The tumors were initially located in the foot, shoulder, and chest wall, respectively, with positive margins at initial resection in wo cases (third case unknown). Metastases were to the lung in all cases. Two children were alive after chemotherapy and one died of disease, in spite of the chemotherapy [16].

The treatment is, first and foremost, surgical. When excision cannot be complete, chemotherapy may be beneficial [16]. Similar to infantile fibrosarcoma, anti-TRK tyrosine kinase inhibitors (larotrectinib, entrectinib, and more recently selitrectinib and repotrectinib) will probably become a standard in the treatment of these tumors.

### 2.4. EWSR1-SMAD3-Rearranged Fibroblastic Tumor

This tumor has been described mainly in young adults, with only two pediatric cases to date. One was a congenital nodule of the foot in a boy [40], the second was a nodule of the hand in a 5-year-old girl, initially diagnosed as “unusual lipofibromatosis” [41]. More cases are needed to determine the epidemiology, clinical features, microscopy, and behavior of this tumor in children.

The tumor seems to have a predilection for acral areas and can develop in the dermis or subcutis. It has an infiltrative growth pattern. In the light of the adult cases, it seems that the tumor typically has a biphasic appearance with a higher cellular density at the periphery and central hyalinization. The spindle cells are arranged in fascicles and contain elongated nuclei with no atypia. Calcifications may be seen in the hyalinized areas. The immunophenotype is characteristic, with constant expression of ERG and negativity for fibroblastic, muscular, and melanocytic markers (CD34, SMA, desmin, caldesmon, S100, and SOX10). The detection of the *EWSR1-SMAD3* (Ewing sarcoma breakpoint region 1–SMA- and MAD-related protein 3) fusion gene is necessary to ensure diagnostic accuracy.

### 2.5. ALK-Rearranged Infantile Fibrosarcoma-like Tumor

Recently, a new *ALK*-rearranged spindle cell tumor has been described in a cohort of four children aged 2 to 10 years. Of these four cases, two cases arose in the superficial soft tissues of the scalp and the hand, respectively. The tumors were both clinically diagnosed as vascular tumors or malformations of about 4 to 6 cm in diameter, however, the histologic appearance was that of a low-grade spindle-cell sarcoma, reminiscent of infantile fibrosarcoma, with some intermixed round cells in one case. The immunoprofile showed negativity for CD34, S100, and SMA, but diffuse positivity for ALK (anaplastic lymphoma kinase). A fusion transcript involving *ALK* was found, with *AK5* and *ERC1* as fusion partners. None of the two tumors recurred and there was no metastasis. However, of the two other cases in this study, which arose in the kidney, one recurred and metastasized to the liver and lung [42]. This child is currently in remission after nephrectomy and chemotherapy based on ifosfamide, doxorubicin, and ceritinib (ALK inhibitor). Therefore, it is possible that this new mesenchymal tumor with *ALK* rearrangement is of intermediate malignant potential. More cases are needed to better understand the prognosis and evolution of this tumor, especially to determine whether cases arising in superficial soft tissues always evolve in a benign manner or might metastasize in the same way that has been described in kidney tumors.

### 2.6. Dermatofibrosarcoma Protuberans/Giant Cell Fibroblastoma

Dermatofibrosarcoma protuberans (DFSP) is one of the most frequent malignant tumors of the skin and superficial soft tissues in children. It remains, however, to be a rare tumor with an annual incidence of about 1 per 1 million [43]. In a recent series of the European Paediatric Soft Tissue Sarcoma Study Group (EpSSG), the median age at diagnosis was 6.9 years [44]. There was no sex predominance; 6.5% of cases were diagnosed before 1 year of age, 58.7% of cases were diagnosed between the ages of 1 and 10, and 34.8% of cases were diagnosed in children over 10 years old. Congenital cases have also been described [45] (Figure 5B). In most series, the trunk is the most common site, followed by the extremities [43]. DFSP is typically a painless, single, slowly growing, reddish or bluish nodule or plaque, usually less than 5 cm in size (Figure 5A). Because of the very slow growth of the tumor the diagnosis is often made very late [45]. Atrophic or morpheaform variants exist [44,46]. Since the tumor is usually small and superficial, imaging is not routinely performed. US, CT scan, and MRI can be used, but the imaging features are not specific [47]. There is, however, a benefit to US or MRI for follow-up after surgery and for recurrent DFSP [48].

In its classical form, DFSP is an ill-defined, invasive tumor, located in the dermis and subcutaneous tissue (Figure 5C). Rarely, the tumor can be primarily subcutaneous with limited dermal involvement. The tumor cells typically infiltrate around adipocytes, resulting in a honeycomb pattern (Figure 5D). DFSP is made of monotonous spindle cells arranged in a storiform pattern (Figure 5E). There is no atypia and no necrosis in the classical form. The mitotic count is low. Different subtypes are recognized in the WHO classification of tumors of soft tissues and bones [8]. The most noteworthy (and very much different microscopically) is giant cell fibroblastoma, consisting of less cellular fascicles of spindle cells, sometimes wavy, in a myxoid or collagenous stroma, and harboring pleomorphic and multinucleated giant cells (Figure 5F) [8,46]. We should also mention (i) pigmented DFSP, also known as Bednar tumor, characterized by the presence of pigmented dendritic cells and (ii) plaque-like DFSP, a more superficial variant that must be distinguished from a plaque-like CD34-positive dermal fibroma [35]. Molecular techniques are helpful, as discussed below. The only useful immunohistochemical marker is CD34, which is positive in the vast majority of cases and usually negative in areas of fibrosarcomatous transformation (Figure 5G) [49,50]. This, along with Ki67 immunostaining, can help in recognizing fibrosarcomatous areas in DFSP, when the typical features (herringbone pattern, hypercellularity, atypical cells, and increased mitotic activity) are not obvious [50]. Such fibrosarcomatous transformation is present in 5 to 16% of adult cases depending on the series [50,51], however, fibrosarcomatous transformation in children remains exceptional with only 12 cases under 21 years of age published in the English literature to date [52].

The hallmark of DFSP in children is the presence of a balanced or unbalanced translocation involving the long arms of chromosomes 17 and 22. The most frequent is the balanced t(17;22) (q21.3;q13.1) translocation, resulting in a *COL1A1-PDGFB* fusion gene (Figure 5H). The unbalanced der(22)t(17;22) is also found in a subset of patients, with possible co-existing balanced and unbalanced translocations in different subclones of the same tumor. Supernumerary ring chromosomes harboring the same t(17;22) translocation seem to be present in adults but not in children [53,54]. This kind of translocations is present in about 90% of cases. In the remaining cases, *COL1A2-PDGFB*, *COL6A3-PDGFD*, and *EMILIN2-PDGFD* fusion transcripts have been identified [55]. A case of DFSP-like tumor with *COL1A1* copy number gain has also been described. In this case, there was not t(17;22) translocation, however, the morphology and immunophenotype were similar to that of DFSP, raising the possibility of a new molecular variant of DFSP [56].

DFSP treatments must achieve the goal of clear surgical margins, which is the only way to avoid recurrences. For this reason, wide local excision is recommended, ideally by Mohs micrographic surgery (MMS). The term “slow-Mohs micrographic surgery” (sMMS) has been used for the FFPE technique (by contrast with immediate frozen section examination), which is a little slower but provides a better morphology and allows the pathologist to use CD34 immunostaining if necessary. For these reasons, the sMMS technique has been preferred by many teams, with the same results as MMS [57,58,59,60]. The surgical margins should be 10 to 11 mm with sMMS and 20 to 40 mm with conventional surgical techniques, or wider in cases of fibrosarcomatous transformation [61,62,63]. After complete resection, the recurrence rate is very low, ensuring a good prognosis of DFSP in children. When a complete resection is impossible, treatment options are limited. DFSP does not respond to conventional chemotherapy and responds moderately to radiotherapy. Targeted therapies (imatinib, sunitinib, sorafenib, pazopanib, everolimus, among others) might be interesting options and the reader is referred to relevant publications on this topic [55,64,65,66,67,68,69,70].

### 2.7. Plexiform Fibrohistiocytic Tumor

Plexiform fibrohistiocytic tumor (PFHT) is a rare neoplasm of intermediate malignant potential with a predilection for children and young adults. The mean age is around 15 but the age range is very wide (from the first to the eighth decade). Originally, a female predominance was described, but this was not obvious in the literature [71]. The most frequent locations are the upper limbs (50%), followed by the lower limbs, and more rarely the trunk and head and neck region. PFHT presents as a slowly growing, painless nodule, usually below 3 cm in greater diameter (Figure 6A,B). Some cases may present as a flat indurated plaque (Figure 6C). Imaging of PFHT is best achieved using MRI and shows either a round mass or a plaque-like lesion, often with infiltrative growth. In about 75% of cases, the tumor is in contact with a bone or tendon. The lesion is typically iso- or hyperintense on T1-weighted imaging, and always hyperintense on T2-weighted imaging with fat suppression. There is enhancement after gadolinium injection [72].

Histologically, PFHT is located in the subcutis and may extend in the dermis (Figure 6D). In most cases, the tumor is poorly circumscribed. Some cases of purely dermal lesions have been described and tend to be better delineated. It is composed of two components: one histiocytic-like component and one fibroblast-like component. These components can be present in variable proportions, thus, defining, classically, three subtypes: fibrohistiocytic, fibroblastic, and mixed [71]. However, in a fibroblastic predominant PFHT, a fibrohistiocytic component should be carefully sought. Indeed, it has been recently stated that purely fibroblastic PFHT might not exist and could actually be reclassified as plexiform myofibroblastoma, a recently described benign neoplasm [35,73]. The characteristic features of PFHT are a plexiform architecture made of nodules of histiocytic cells with bland cytology, frequently admixed with osteoclast-like giant cells, and intersecting bundles of fibroblast-like spindle cells at the periphery of the nodules (Figure 6E–G). There is little to no pleomorphism and the mitotic activity is low. Iron deposition is common (Figure 6H). Some atypical cases have been reported, with pleomorphism, atypical mitoses, or even intravascular invasion, but their significance remains unclear. Immunohistochemistry reveals a positivity for CD68 and CD163 (personal data, Figure 6I) in the histiocytic-like and osteoclast-like giant cells and often some positivity for SMA in the fibroblast-like cells (Figure 6J) [71].

From a histological point of view, the differential diagnoses of PFHT in children are those of plexiform (or plexiform-like) pediatric soft tissue tumors, mainly: fibrous hamartoma of infancy, plexiform neurofibroma (in the context of neurofibromatosis type 1), cellular neurothekeoma (a benign, poorly circumscribed, dermal tumor, showing CD10 and NK1C3 reactivity), and the already mentioned plexiform myofibroblastoma. Interestingly, cellular neurothekeoma may belong to the same spectrum as PFHT, the former being a superficial benign variant of the later [74]. On a small biopsy, PFHT may also be misdiagnosed as a granulomatous lesion [75].

The prognosis of PFHT is usually favorable, with possible local recurrence, more rarely multifocal, and exceptional lymph node or lung metastases. Even in metastatic cases, the outcome was favorable in nearly all reported cases, except for one. At present, there are no known parameters, either histological, cytogenetic, or molecular, to predict the evolution of the tumor. No genetic anomaly has been described so far [75]. Therefore, the treatment of choice is the complete excision of the tumor.

## 3. Vascular Tumors

### 3.1. Kaposiform Hemangioendothelioma

The incidence of kaposiform hemangioendothelioma (KHE) is probably under evaluated. It has been suggested that small asymptomatic or atypical KHE may be misdiagnosed as variants of other vascular tumors [76]. Although the exact incidence and prevalence of this vascular tumor are not known, it is clear that there is a peak within the first year of life. Around 50% of superficial cases are congenital [76]. It is admitted now that KHE and tufted angioma belong to the same spectrum and that they share the same biology and probably the same pathogenesis [77,78]. Actually, some specialists argue that KHE might only be a florid presentation of tufted angioma, and that the adverse clinical course of this tumor is purely dependent on the occurrence of Kasabach–Merritt phenomenon (KMP) or vital organ compression. For this reason, KHE may well be benign, the morbidity and mortality being linked to secondary complications. We decided to include KHE in this review anyway, but this decision is debatable and it will not be discussed extensively.

KHE usually involves the deep dermis and subcutis, but deeper lesions are possible, with no cutaneous signs in about 12% of cases [76,78]. There is a slightly higher prevalence in the limbs, but all sites are possible. Most of the time, the tumor is unique, and can adopt many clinical aspects: erythematous sometimes purple; papules, nodules or plaques or an indurated mass; with varying degrees of infiltration [78]. It is a big, rapidly growing tumor, 3 to 27 cm in greater dimension [79]. One of the most severe complications of KHE, also seen in tufted angioma, is the KMP, a life-threatening consumptive coagulopathy with severe thrombocytopenia. It occurs in 42 to 71% of cases [80,81,82]. There is a higher frequency of KMP in congenital, large KHE, especially above 8 cm [83]. In cases of KMP, the KHE lesions appear purpuric, congestive, and painful [84]. Spontaneous hemorrhage is rare but any invasive procedure (biopsy, excision), trauma, or ulceration may lead to significant bleeding. Other complications of KHE include musculoskeletal disorders due to tumor infiltration, lymphedema, and compression of vital structures (e.g., airway obstruction in a KHE of the neck) [78].

US reveals an ill-defined, heterogeneous lesion, most often hyperechoic [79]. On color Doppler US, the tumor is hypervascular. MRI reveals a heterogeneous, hyperintense appearance on T2-weighted images, with heterogeneous, generally intense enhancement [79]. Overall, the imaging findings lack specificity and are characterized by a wide range of aspects regarding the degree of infiltration, signal intensities, and enhancement patterns [79].

The gold standard for diagnosis is the biopsy. KHE is a deep dermal and subcutaneous, infiltrating proliferation of spindle endothelial cells arranged in nodules (Figure 7A). The tumor can adopt a “cannonball” pattern, especially in the dermis, similar to a tufted angioma, but the nodules are larger, less defined, and coalescent. The tumor is also more infiltrative. Mitoses are rare and not abnormal (Figure 7B) [78,80,85]. Most of the spindle cells are positive for lymphatic markers podoplanin, LYVE1, and PROX1 (Figure 7C) [80,86].

Similar to other vascular anomalies, numerous mutations involving G proteins have been found in KHE and tufted angioma, especially in the *GNA14* gene, a paralogue of *GNAQ* [87]. These genes encode the Gα subunit that, along with Gβ and Gγ in a heterotrimer, binds G protein-coupled receptors.

The prognosis depends on the size and location of the tumor. Large, deep KHE tumors are associated with higher morbidity and mortality due to tumor infiltration, compression of vital organs, and KMP [78]. Treatment with corticosteroids and/or vincristine has been recommended but is based on expert opinion and lack a well-designed clinical study [84]. Wide surgical excision can be curative but is rarely feasible. For a decade now, sirolimus has been emerging as a treatment option. The first report of a successful treatment in a child was published in 2010 [88]. Some of the concerns with sirolimus have been the occurrence of adverse events and the issue of the right dose in the pediatric population [89]. Recently, recommendations for the initial dose in children were proposed. Due to sirolimus being metabolized by cytochrome P450 3A (CYP3A), the initial dose depends both on the weight and the *CYP3A5* genotype [90]. A recent prospective study of sirolimus for complicated vascular anomalies underlined the importance of closely monitoring possible adverse events, while confirming the effectiveness of the treatment [91]. In this study, the most frequent adverse events (>20% of cases) were mucositis, upper respiratory infections, and nausea/vomiting.

### 3.2. Papillary Intralymphatic Angioendothelioma (PILA)/Retiform Hemangioendothelioma

Papillary intralymphatic angioendothelioma (PILA, also known as Dabska tumor) and retiform hemangioendothelioma belong to the same spectrum. These two rare tumors are lymphatic in origin. Cases with overlapping features of PILA and retiform hemangioendothelioma have been described, which is the reason why some authors have suggested to include both tumors under the name *hobnail hemangioendothelioma* [92]. However, pure forms of retiform hemangioendothelioma are very rare in children; PILA or mixed tumors are much more frequent. PILA usually involves the dermis and subcutaneous tissues of the head and neck in children and adolescents, whereas retiform hemangioendothelioma more often involves the upper and lower limbs of young adults and adolescents, with some reported cases in the head or trunk [93]. Similar to Masson’s tumor, PILA has been reported to arise in a pre-existing vascular malformation, but also on chronic lymphedema and in intramuscular hemangiomas. There are a few reports of a retiform hemangioendothelioma arising on a pre-existing lymphatic malformation [94].

PILA and retiform hemangioendothelioma both present as a slowly growing, firm, solitary nodule or mass, more rarely plaque; all nonspecific findings, which explains the diagnosis not being raised by dermatologists or pediatricians. To the best of our knowledge, there has been no study of the imaging characteristics of these tumors in the English literature.

Microscopically, PILA consists of vessels of various sizes, more or less dilated, containing papillary tufts protruding from the vessel wall and sometimes occluding the lumen. The papillae are centered by a hyalinized core and lined by hobnail endothelial cells with large hyperchromatic nuclei located towards the luminal pole of the cell (Figure 8A). Mitoses are sparse. Retiform hemangioendothelioma involves the whole dermis and often infiltrates the subcutaneous tissue. It is made of elongated, anastomosing vessels reminiscent of the architecture of the rete testis. In both PILA and retiform hemangioendothelioma, there is often a prominent lymphocytic infiltrate. The immunophenotype is similar in both tumors, with positivity for CD31 and CD34 in the endothelial cells. Prox1 and podoplanin (D2-40) are positive in most cases of PILA (Figure 8B) [95,96,97,98]. Of note, rare cases of composite hemangioendothelioma have been described in adolescents, consisting of areas of retiform hemangioendothelioma and solid areas with a vaguely neuroendocrine morphology. These cases showed no recurrence nor metastasis in children, however, both of these have been reported in adults, especially bone and lung metastases [96]. PILA has intermediate malignant potential with possible local recurrences and exceptional lymph node metastases. Lung metastases have been described in at least one case [99]. Retiform hemangioendothelioma shares a similar clinical course.

The treatment of choice of PILA and retiform hemangioendothelioma is complete surgical resection, in order to avoid recurrences. However, clear surgical margins may be hard to achieve in some cases of retiform hemangioendothelioma, due to its infiltrative growth pattern. In such cases, MMS may be useful, as reported in a 11-year-old girl with a tumor of the finger [100]. In unresectable cases, association of radiation therapy and chemotherapy with cisplatin might be an option, as reported in an adult [101].

### 3.3. Pseudomyogenic Haemangioendothelioma

Pseudomyogenic hemangioendothelioma (PMHE) is a rare vascular tumor of intermediate malignant potential. It affects adolescents and young adults, with a mean age at diagnosis close to 30 years and a strong male predominance (sex ratio about 4:1) [102,103]. The most frequently involved sites are the extremities, especially the lower limbs, followed by the trunk and upper limbs, but the tumor has been described in other locations. PMHE arises mainly in the soft tissues but may occur in the dermis, muscle, or bone [102]. A characteristic feature of this tumor is the frequent involvement of different tissue planes, resulting in a mass or deep-seated nodule, with pain in about half of cases. Multicentric presentation is also common [102,104,105]. On MRI, PMHE is hypointense on T1-weighted images and hyperintense on T2-weighted and stir-weighted images. It has been suggested that positron emission tomography scan (PET-scan) might be helpful to reveal possible occult deep lesions [8].

Grossly, PMHE is usually made of multifocal, ill-defined, 1 to 3 cm large, white- or brown-colored nodules [105]. Microscopically, PMHE has an infiltrative pattern of growth (Figure 9A). It is made of sheets and fascicles of epithelioid cells or plump spindle cells with a deeply eosinophilic cytoplasm which can be reminiscent of rhabdomyoblasts (Figure 9B). The stroma is loose, rarely myxoid, and contains numerous neutrophils in about half of cases (Figure 9C) [105]. Similar to dermatofibroma, epidermal hyperplasia may be present above the tumor [106]. Intravascular invasion has been described but does not seem to have any prognostic impact [107]. PMHE co-expresses cytokeratins and endothelial markers. It is diffusely positive for cytokeratin AE1/AE3 and ERG. CD31 is positive in about half of cases. SMA may be expressed focally in some cases. Interestingly, there is diffuse nuclear positivity for FOSB in 96% of cases, a very helpful finding since FOSB is expressed in only a couple of other vascular tumors: epithelioid hemangioma (54% of cases), which is not a histological differential diagnosis of PMHE, and some rare cases of epithelioid angiosarcoma (5%), spindle-cell angiosarcoma (10%), or epithelioid hemangioendothelioma (5%) [104]. One of the main differential diagnoses of PMHE is epithelioid sarcoma. However, INI1 is normally expressed in PMHE.

FOSB positivity is related to the presence of the t(7;19)(q22;q13) translocation responsible for a *SERPINE1-FOSB* gene fusion in the majority of PMHE [102,108]. Recently, other partners for *FOSB* have been found in some cases of PMHE: the beta-actin gene, responsible for the *ACTB-FOSB* gene fusion [109,110]; the WW domain-containing transcription regulator 1 gene, responsible for the *WWTR1-FOSB* gene fusion in an isolated case [111]; and more recently, the clathrin heavy chain gene, responsible for the *CLTC-FOSB* gene fusion [112]; and the epidermal growth factor-like 7 gene, responsible for the *EGFL7-FOSB* gene fusion [113]. It remains to be determined if these various gene fusions are associated with different clinicopathological features. To date, there is no targeted therapy for *FOSB* gene fusions.

The clinical course is characterized by local recurrences in about 60% of cases, often multiple and with possible additional nodules. Regional lymph node metastasis has been described in only one patient, and distant metastases in three patients; only two patients died of disease [107,114,115]. The treatment of choice is complete surgical excision. Some reports of clinical improvement following treatment with sirolimus or everolimus (anti-mTOR) are hopeful for non resectable cases [116,117].

### 3.4. Angiosarcoma

Angiosarcoma is extremely rare in children, therefore, it is impossible to draw reliable epidemiological or clinical data on angiosarcoma in children and in the skin. In children, it is mostly seen in association with genetic conditions such as xeroderma pigmentosum, or Aicardi syndrome, or after radiation therapy [85]. Therefore, angiosarcomas are almost always seen in older children and adolescents. Congenital or infantile cases are exceedingly rare. Angiosarcomas in children seem to predominate in girls and in the lower extremities, but these observations should be tempered since they are based on a very small series of 10 cases [118]. The tumor is often a rapidly enlarging nodule or mass, sometimes ulcerated (Figure 10A). It may be centered in the dermis, in the subcutaneous tissue, or in both. The tumor is made of a branching network of vessels admixed with more solid areas made of epithelioid cells (Figure 10B). Indeed, the epithelioid variant is particularly frequent in children (90%). Nuclear atypia is often prominent, at least in some areas. Mitotic activity is high in most cases (Figure 10G), but may be low. Necrosis is possible. Immunohistochemistry reveals a positivity for at least one endothelial marker, with a better sensitivity of CD31 and ERG as compared with CD34 (Figure 10C–E) [118]. Podoplanin is commonly positive. HHV8 is negative. Aberrant cytokeratin positivity is possible (Figure 10F). Wide surgical excision is the treatment of choice. Adjuvant chemotherapy may be useful but reported cases are too few to draw any recommendation and each case should be discussed collegially.

### 3.5. Kaposi Sarcoma

Kaposi sarcoma (KS) is a low-grade malignant vascular neoplasm caused by human herpesvirus 8 (HHV8), also known as KS-associated herpesvirus. Four distinct clinical subtypes are usually recognized: classic indolent (sometimes called “Mediterranean”) KS, African/endemic KS, AIDS-associated KS, and iatrogenic KS. The classic indolent form is exceedingly rare in children, even in families with a genetic predisposition, where it usually occurs in the fifth decade or late fourth decade [119]. The three other subtypes are a little more frequent, but KS remains a rare tumor and there is striking geographical disparity. African/endemic KS occurs in both adults and children, especially in equatorial Africa. In Western countries, it is an imported disease, and the diagnosis should be raised in immigrant children. AIDS-associated KS, which is the most aggressive type, occurs through mother-to-child contamination, however, the risk has been reduced with the advent of antiretroviral therapies. Iatrogenic KS, also very rare, seems to have become more frequent due to advances in solid-organ transplantation [8]. The clinical presentation differs among subtypes and there are some pediatric particularities. Endemic KS in children has a clear tendency to present as multiple lymphadenopathies with only rare skin lesions, a rapidly progressive course, and a high mortality [120]. AIDS-associated KS presents with typical purplish, reddish, or brown plaques or nodules that may ulcerate, sometimes with an anemic rim which is very telling when present. The lesions are mostly found on the face, genitals, and lower limbs, with frequent involvement of oral mucosa, digestive tract, lungs, and lymph nodes. Of note, a case of immune reconstitution inflammatory syndrome (IRIS)-associated KS has been described in a child following introduction of antiretroviral therapy [121]. Iatrogenic KS may develop within a few months to several years after solid-organ transplantation or immunosuppressive therapy for other conditions. The clinical course is fairly unpredictable and some cases may regress upon withdrawal of the immunosuppressive treatment [8]. KS can also be associated with congenital immunodeficiency syndromes, such as Wiscott–Aldrich syndrome (Figure 11A) [122]. Such cases are very rare but may be seen by pediatric dermatologists, especially in pediatric hospitals caring for these children.

Microscopically, all four subtypes are identical. For the most part, KS is easy to recognize: it is made of a vascular proliferation dissecting or replacing collagen fibers in the dermis, surrounded by a more or less prominent spindle-cell proliferation of endothelial cells admixed with a dense infiltrate of lymphocytes and plasma cells, often accompanied with extravascular erythrocytes and hemosiderin deposits. However, the aspect depends on the stage of the lesion (Figure 11B–E). One of the peculiarities of KS is the very wide range of morphologies that the tumor can adopt [123]. The most specific immunostaining is nuclear HHV8. In rare negative cases, the diagnosis may be confirmed by PCR.

The treatment is adapted to the clinical subtype but also depends on available drugs and facilities. It includes surgery, radiation therapy, chemotherapy, and of course, antiretroviral drugs in AIDS-associated KS. Treatment recommendations in children are effectively summarized in a paper by Molyneux et al. [124].

## 4. Muscular Tumors: Rhabdomyosarcomas

Primary cutaneous rhabdomyosarcoma (RMS) is a rare tumor, with about 55 cases reported in the literature [125,126]. It occurs mainly in the head and neck, and there seem to be no sex predominance [125]. The age range is wide with rare congenital/neonatal cases, but also cases in older children and adolescents. Importantly, the two main histological subtypes of RMS, embryonal RMS (ERMS) and alveolar RMS (ARMS), have distinct clinical and histological characteristics. Therefore, they are discussed separately hereafter. Contrary to non-cutaneous RMS, where ERMS account for nearly 75% of all RMS, primarily cutaneous RMS are more often of the alveolar subtype [125], even in congenital/neonatal cases. Indeed, in the skin and superficial soft tissues, after a short review of the literature, we found that congenital/neonatal ARMS were twice as often represented as congenital/neonatal ERMS (20 ARMS, 9 ERMS) [127,128,129,130,131,132]. In children, tumors with extensive pleomorphism should be considered anaplastic ERMS and not pleomorphic RMS [133]. To the best of our knowledge, no pediatric primary cutaneous spindle-cell/sclerosing RMS has been reported in the literature. Still, it seems important to keep in mind this differential diagnosis when confronted with a spindle-cell tumor, since this RMS subtype is associated with a better prognosis and indolent clinical course. *NCOA2* and *VGLL2* rearrangements are the most common in spindle-cell/sclerosing RMS in children and should be sought for establishing this diagnosis [134].

### 4.1. Alveolar Rhabdomyosarcoma (ARMS)

Congenital/neonatal ARMS has a slight female predominance [130,135]. Although very rare in this age group, congenital/neonatal ARMS has other particularities which are worth mentioning: the tumor occurs mainly in the limbs, followed by the head and neck, and the trunk [130]; according to a recent review, cutaneous and subcutaneous locations are most often metastatic sites [130,136] however, at least four reported cases of congenital ARMS were primarily cutaneous [128,131,135,137]. All ages considered, ARMS is usually an erythematous mass or nodule, initially misdiagnosed as a hemangioma, but often rapidly growing, ranging from 4 to 12 cm in greater diameter [128,131,137]. When congenital/neonatal, it may be multiple, even in primarily cutaneous cases. In older children, the lesion is unique (Figure 12).

Microscopically, primarily cutaneous ARMS is similar to ARMS in other locations; it is composed of nests of undifferentiated small round cells with a high N/C ratio and a variable proportion of tumor cells with a moderate eosinophilic cytoplasm. True differentiated rhabdomyoblasts are rare. In the center of the nests, the cells show discohesion with a tendency to disaggregate, leading to the alveolar pattern. Multinucleated tumor cells are a rather common finding. Mitoses can be sparse or more prominent and necrosis is possible [127,128,132]. Tumor cells express skeletal muscle markers: desmin, myogenin (Myf4), and/or MyoD1.

The overall prognosis of congenital ARMS is poor, with about 75% of deaths in the review by Russo et al. However, about 75% of cases in this review also had metastases, with 41% of central nervous system metastases [130]. It is possible that primary cutaneous cases of congenital ARMS have a better prognosis, as reported [128,131,135,137]. Importantly, the prognosis of RMS, whatever the age of the child, has been linked to its genetic anomalies; patients with *FOXO1* fusion positive RMS having a worse outcome than patients with *FOXO1* fusion negative RMS, and the *PAX7-FOXO1* fusion has a better prognosis than the *PAX3-FOXO1* fusion [138]. Since 70% of ARMS show either one of the *FOXO1* fusions, it makes sense that ARMS should have a poorer prognosis than ERMS. However, in congenital/neonatal ARMS cases, only three *FOXO1* fusions have been reported, to our knowledge, whereas the majority of cases seem to be negative for *FOXO1* fusions [127,129,130,135]. This suggests a possible different pathogenesis for congenital/neonatal ARMS. For instance, some cases have been linked to Beckwith–Wiedemann syndrome and the tumors seem to arise from silencing of the *CDKN1C* gene through epigenetic mechanisms in this syndrome [129,135].

### 4.2. Embryonal Rhabdomyosarcoma (ERMS)

Congenital/neonatal ERMS mainly occurs in the head and neck, followed by the trunk. The male/female ratio is about two [130]. Only nine cases of congenital/neonatal ERMS of the skin or superficial soft tissues have been reported, to the best of our knowledge, with fragmentary data, which makes it difficult to draw meaningful conclusions [127,139,140,141]. The locations include the cheek, scalp, neck, hand, foot, and lower back. All ages considered, the tumor is firm, ranges from 3 to 15 cm in greater dimension, may be ulcerated if large, with possible associated lymphadenopathy [127,139,141].

Microscopically, ERMS shows alternating loose and dense cellularity in a variably myxoid stroma. The tumor cells range from stellate to small round undifferentiated to spindle. Unlike ARMS, myogenin expression is more variable.

The prognosis is deemed to be better than that of ARMS, however, in congenital/neonatal cases at least three out of nine ERMS cases were fatal after only a few months: the tumor recurred in one case in spite of surgery, chemotherapy (specific RMS protocol), and radiation therapy; the tumor recurred in one other case after surgery; and one child did not receive any treatment [127,139,141]. The outcome was not known or favorable in other cases. Interestingly, one case exhibited a t(2;8)(q35;q13) translocation. When it was performed, cytogenetic techniques never evidenced any *FOXO1* fusion. One case was associated with Apert syndrome (craniosynostosis, midface hypoplasia, and syndactyly of the hands and feet), a syndrome driven by mutations in *FGFR2*. In a recent study of germline predisposition variants in pediatric RMS, the authors found some cancer predisposition gene variants in ERMS occurring before 1 year of age, but not in ARMS in this age group. The exact age and location of the tumors are not discussed in the paper, but the most frequently involved genes have been *NF1* (neurofibromatosis type 1), *TP53* (Li–Fraumeni syndrome), *HRAS* (Costello syndrome), *CBL* (Noonan syndrome), and *BRCA2* [142]. It is worth mentioning that some cases of congenital/neonatal ERMS may arise on congenital melanocytic nevi [141].

### 4.3. Rhabdomyosarcoma Risk Group Stratification and Treatment

The outcome and treatment of RMS are dependent on risk groups. Of note, the perineal location seems to have prognostic significance as well, with a poorer prognosis in those cases [126].

The stratification of patients into risk groups at diagnosis is essential. The definition of risk groups is different in Europe and in North America, with different treatment strategies. In Europe, four risk groups (low, standard, high, and very high) have been defined by the European Paediatric Soft Tissue Sarcoma Group (EpSSG), based on the Intergroup Rhabdomyosarcoma Study Group (IRS) post-surgical stage, the age, the tumor size, the histopathological subtype, the site of the primary tumor, and the nodal stage. According to the EpSSG, surgical resection follows induction chemotherapy when any residual mass can be completely excised (R0/R1) without causing a significant organ or functional impairment. For some standard risk cases, radiation therapy may be avoided if the resection is complete (R0). For more details on the management of RMS and the differences between North American and European approaches, the reader is referred to the review by Yechieli et al. [143].

## 5. Adipocytic Tumors: Liposarcomas

Liposarcomas of the dermis and subcutaneous tissue (also named superficial liposarcomas) are exceedingly rare in children. They occur almost exclusively in adolescents, and it remains unclear if boys or girls are more affected [144,145]. Nonetheless, in two series of superficial liposarcomas, cases were described in patients as young as 5 and 6 years old [146,147]. In a series of pediatric cases, there was a strong predilection for the lower limbs (~60%), followed by the head and neck, and upper limbs [144]. The tumors were quite large, with about 50% of cases >5 cm. Lymph node involvement was present in 6% of cases and metastases in only one case (3%). Given the rarity of both superficial liposarcoma and pediatric liposarcoma, they are discussed in detail.

Three types of superficial liposarcomas are recognized in children, by order of frequency: myxoid liposarcoma, pleomorphic liposarcoma, and atypical lipomatous tumor (a term preferred to “well-differentiated/dedifferentiated liposarcoma” in superficial locations, given the indolent course and favorable prognosis of this tumor).

The diagnosis of myxoid liposarcoma is confirmed by the presence of the t(12;16)(q13;p11) translocation, resulting in a *FUS-DDIT3* gene fusion. *EWSR1-DDIT3* fusions have also been reported.

It is unclear if pleomorphic liposarcoma per se really exists in children. Indeed, a “myxoid pleomorphic liposarcoma” subtype, with a predilection for children and poor prognosis, has recently been added to the WHO classification of tumors of soft tissues [8]. Thus, previously reported cases of pleomorphic liposarcomas in children might have been cases of myxoid pleomorphic liposarcomas. However, to the best of our knowledge, this new subtype has never been described in the superficial soft tissues.

The prognosis depends on the histologic subtype. Myxoid liposarcoma has an excellent prognosis, myxoid pleomorphic liposarcoma has a poor prognosis, and atypical lipomatous tumor has an indolent behavior. Wide local excision is the treatment of choice. Of note, liposarcomas in children may be associated with predisposition syndromes such as Aicardi syndrome, Li–Fraumeni syndrome, Muir–Torre syndrome, PTEN hamartoma tumor syndromes, probably most lipomatoses, and syndromes belonging to the *PIK3CA*-related overgrowth spectrum [145].

## 6. Neuroectodermal and Neural Tumors

### 6.1. Melanotic Neuroectodermal Tumor of Infancy

Melanotic neuroectodermal tumor of infancy (MNTI) is a rare tumor occurring mostly in infants under 1 year of age. Various names have been used for this tumor, including melanotic progonoma, retinal anlage tumor, pigmented ameloblastoma, melanotic adamantinoma, melanotic epithelial odontoma, pigmented congenital epulis, pigmented teratoma. The designation MNTI is retained in the WHO classification of tumors of soft tissues and bone [8]. Although its pathogenesis is not understood, it is established that MNTI derives from neural crest cells. Recently, DNA methylation profiling demonstrated a common signature between MNTI, some medulloblastomas, and possibly pineal anlage tumor [148]. MNTI is rare in the skin and superficial soft tissues, with less than 10 cases reported in the English literature [149,150,151]. However, our experience is that this tumor may be more frequent than reported.

Most cutaneous cases are located on the limbs and present as a non-tender, firm swelling. The tumor can grow rapidly but is not adherent to the overlying skin and most often mobile. The skin can be erythematous or show prominent telangiectasia. The tumor is often quite large, between 2 and 6 cm in greater dimension [149,150,151,152]. The typical bluish-black color of the tumor is not always visible on dermatological examination, but it is a striking finding on sampling of the tumor, which is better performed by incisional biopsy or direct excision surgery when possible [149,150]. The 24-hour urinary vanillylmandelic acid levels can be elevated in some cases [151]. Imaging findings are not specific, unless involvement of an underlying bone is seen, which may trigger suspicion of MNTI, but also of Ewing sarcoma or another bone tumor extending to the superficial soft tissues [151].

Microscopically, MNTI is typically composed of a biphasic population of small, neuroblast-like or undifferentiated cells and larger, epithelioid, melanin-producing cells. The proliferation is arranged in nests, cords, or trabeculae in a fibrous, sometimes desmoplastic stroma. Often, the pigmented epithelioid cells surround the small undifferentiated cells (Figure 13A–D). In some cases, only the small blue cells are seen, which makes the differential diagnosis with an Ewing sarcoma or a neuroblastoma metastasis challenging [149,150,151,152,153]. Immunohistochemistry is not necessary for the diagnosis in typical cases and, when performed, is often inconsistent among different cases. The melanin-producing epithelioid cells are usually pancytokeratin (PanCK) and HMB45 positive, whereas the small blue cells are usually synaptophysin positive (Figure 13E,F) [150]. Of note, rare membranous expression of CD99 as well as focal rhabdomyoblastic and glial differentiation have been described in MNTI of the craniofacial bones, however, to our knowledge, it has never been described in MNTI of the soft tissues [150,153].

As of today, a *BRAF*V600E mutation has been described in one MNTI, and a case of *CDKN2A* germline mutation too [154,155].

Superficial soft tissue cases of MNTI have an overall good prognosis with five patients alive without recurrence or metastasis (with a 3 to 40 months follow-up) out of eight cases. Follow-up was not available for two cases. Only one patient had metastases and died of disease [149,150,151]. Interestingly, this was the only case treated only with chemotherapy and radiation therapy. In all other cases, surgical excision of the tumor was performed (after induction neoadjuvant chemotherapy in one case) and was sufficient. It has been suggested that the “debulking effect” of surgery might be a trigger for body antitumor defenses and subsequent tumor regression [156].

### 6.2. Ewing Sarcoma Family of Tumors

Ewing sarcomas and associated tumors belonging to the Ewing sarcoma family are exceedingly rare in the superficial soft tissues and the skin. Less than 150 cases of primary superficial Ewing sarcoma have been reported in the literature. Ewing sarcoma usually arises in bone, with about 15% of cases arising in deep soft tissues and even less in superficial soft tissues and skin. Superficial Ewing sarcoma (sES) shows a female predominance and the median age ranges from 17 to 21.5 years depending on the series [157]. There has been reported cases in younger children, or even congenital cases [158,159]. Considering both children and adults, the most common primary sites are the trunk and lower limbs, followed by the upper limbs and head and neck. However, as far as children are concerned, the distribution pattern is less clear. The tumor’s median size is about 3.5 cm in greater diameter. Fever may be a symptom in 8.5% of cases. Metastases at diagnosis are rare (2 to 3.5% of cases), the lung being the most frequent site [157]. Microscopically, sES are identical to their deep counterparts: they are made of sheets of small round blue undifferentiated cells with fine chromatin and inconspicuous nucleoli, sometimes with PAS-positive cytoplasmic granulations, and with a very sensitive diffuse membranous CD99 staining pattern. Importantly, CD99 is not specific and may be positive in many other tumors: synovial sarcoma, acute lymphoblastic leukemia, melanoma, solitary fibrous tumor, *BCOR*-rearranged sarcoma, angiomatoid fibrous histiocytoma, neurothekeoma, and calcifying aponeurotic fibroma. This list is not exhaustive. As a consequence, the diagnosis needs to be validated molecularly. The most frequent translocation is t(11;22)(q24;q12) resulting in the *EWSR1-FLI1* gene fusion [160]. The second most common translocation is t(21;22)(q22;q12) resulting in the *EWSR1-ERG* gene fusion [161]. Importantly, ERG immunostaining is positive in these cases [162]. Other rare fusions involve *ETV1*, *ETV4*, and *FEV*, but also *FUS* in place of *EWSR1*, resulting in *FUS-ERG* or *FUS-FEV* gene fusions [163]. Given the rarity of sES, there is currently no specific recommendation for treatment. However, sES has a more indolent course than its bone counterpart [164]. Metastatic disease is associated with a worse prognosis. In addition, metastatic relapse after surgery for localized disease has been reported. Therefore, chemotherapy has an important place in the treatment of sES, and it has been stated by some authors that the outcome might be better with pre-operative chemotherapy, since nearly 50% of primary surgeries were intralesional or marginal in their series [157].

### 6.3. Malignant Peripheral Nerve Sheath Tumor (Associated with Neurofibromatosis Type 1)

A few cases of malignant peripheral nerve sheath tumors (MPNST) have been reported in adolescents with neurofibromatosis type 1 (NF1). However, a diagnosis of MPNST in a child without NF1 should be considered with the utmost circumspection, unless previous irradiation exists. NF1 is an autosomal dominant genetic disease with an estimated incidence between 1/2500 and 1/3000 individuals. The *NF1* gene, located at 17q11.2, encodes the protein neurofibromin, whose tumor suppressor function consists in the downregulation of the RAS-RAF-MEK pathway [165]. Clinical criteria for the diagnosis of NF1 have been issued by the NIH and include multiple café-au-lait macules (CALM), skin-fold freckling, Lisch nodules, optic pathway gliomas, distinctive bone lesions such as long-bone dysplasia and sphenoid wing dysplasia, and neurofibromas. However, most of these signs occur in late childhood and CALM are often the only diagnostic feature in young children. Therefore, revised diagnostic criteria have been suggested [166,167]. The most important new diagnostic criterion is the detection of a pathogenic *NF1* variant, and this is especially useful in distinguishing NF1 from Legius syndrome or constitutive mismatch repair deficiency (CMMRD) [167].

Patients with NF1 have from 8% to 12% lifetime risk of developing a MPNST. The tumor is often deep seated and a significant proportion arises on a pre-existing neurofibroma. Superficial cases are even rarer. In patients with NF1, who often harbor numerous neurofibromas, warning signs are a rapidly growing mass, pain, and neurologic deficit [168]. MRI imaging of MPNST reveals an ill-defined, soft tissue mass, typically over 4 cm in greater dimension, without contiguity with adjacent nerves. The tumor is heterogeneous with possible intralesional hemorrhage and an infiltrative margin on T1-weighted images. There is no target sign on T2-weighted images, but possible intra-tumoral cysts and perilesional edema. Heterogeneous and perilesional enhancement are present after gadolinium injection. 18F-FDG-PET imaging is useful in differentiating an MPNST from a benign neurofibroma; MPNST typically shows maximal SUV over 3.5 and a tumor-to-liver ratio >2.6, with high sensitivity and specificity [169].

Microscopically, MPNST may adopt very diverse appearances. They often show a herringbone, fibrosarcoma-like architecture, or more haphazardly oriented fascicles. The tumor is densely cellular, with mild nuclear atypia. Mitotic activity is often elevated (Figure 14A,B). Necrosis is frequent. A divergent differentiation is possible (malignant triton tumor), such as cartilage, bone, skeletal muscle, or smooth muscle. Epithelioid MPNST represent less than 5% of all cases. Some cases are difficult to classify, with features intermediate between MPNST and cellular neurofibroma. There is currently no widely recognized recommendation for the diagnosis of such cases, termed “atypical neurofibromatosis neoplasms of uncertain biologic potential” (ANNUBP). Immunohistochemistry is useful in assessing the transformation of a neurofibroma into an MPNST. In MPNST, S100, SOX10, and H3K27me3 are often lost (Figure 14C); p16/CDKN2A is also lost (but this is also the case in some ANNUBP); TP53 may be diffusely positive in high-grade cases; and Ki67 is greater than 10% [170].

The surveillance of NF1 children is probably the most important item in patient care. Annual history and physical exam are recommended, including extensive skin and neurological exams, ophthalmologic assessment by a specialist, and search for signs of MPNST (large tumor, pain, rapid growth, and neurologic deficit) with special attention for previously irradiated body regions, if relevant [168]. Atypical neurofibroma and ANNUBP should be surgically resected, with preservation of neurological function whenever possible. The management of MPNST in children is not well defined and should be discussed in multidisciplinary meetings on a case-by-case basis. There is no consensus on the use of chemotherapy. Radiation therapy can be used before surgical excision in large, high-grade MPNST, when a small radiation field can be applied, which is often the case in superficial tumors [170]. MPNST is an aggressive tumor with poor prognosis, and there seem to be a decreased survival rate in NF1 patients.

## 7. Tumors of Unknown Differentiation

### 7.1. Epithelioid Sarcoma

Epithelioid sarcoma (EpS) is a rare tumor mainly seen in adolescents, with about 50% of cases occurring between the ages of 15 and 19, according to the review by Liszewski from the SEER-18 database. However, it can be seen in younger children and about 1.6% of cases can occur congenitally [1]. In a recent prospective analysis from the Children’s Oncology Group (COG) and the European Paediatric Soft Tissue Sarcoma Study Group (EpSSG), the median age was 13, a little younger than found by Liszewski et al. [171]. There seems to be a slight male predominance. Half of cases occur in the upper limbs, about 20% in the lower limbs, followed by the head and neck, and the trunk. Most cases in children are distally located (80%) and are also known as “classic” EpS, as opposed to the proximal variant of the tumor [171]. Clinically, classic EpS is a solitary, slow-growing, firm and painless nodule. Multiple nodules are also possible. Ulceration is quite frequent. Proximal EpS is often multinodular and deep seated. The tumor size varies from 0.4 to 19 cm in diameter. About 14% of cases present with lymph node metastasis [171].

A diagnosis of EpS is based on pathology. EpS is located in the subcutaneous tissue in about 76% of cases and rarely occurs primarily in the dermis (about 4% of cases) [1]. Classic EpS is characterized by irregular nodules of epithelioid cells, often poorly cohesive and with a spindle peripheral component in a desmoplastic stroma. Central necrosis can be seen in the nodules and, if confluent, may result in large geographical necrosis areas, which can mimic a necrobiotic granulomatous lesion. The cells have deeply eosinophilic cytoplasm (Figure 15A–C). Mitoses are sparse [172]. Histological variants have been described, most importantly the proximal-type variant. In proximal EpS, the tumor is prominently multinodular, made of large, rhabdoid cells. Foci of necrosis are frequent, but the pseudo granulomatous pattern of classic EpS is absent. Classic and proximal EpS are both positive for cytokeratins and EMA. CD34 is positive in about 50% of cases, as well as ERG, which should be known to avoid confusion with a vascular tumor, especially in the angiomatoid variant. CD31 and FLI-1 are usually negative [172]. The major immunohistochemical characteristic of EpS is the loss of nuclear expression of SMARCB1 (INI1, BAF47) (Figure 15D) [173]. Indeed, EpS is associated with loss of SMARCB1 expression, mostly through biallelic deletions involving the *SMARCB1* gene, more rarely through monoallelic deletions or microRNAs interactions with *SMARCB1* transcripts [174,175]. Other SWI/SNF chromatin-remodeling complex members are involved in EpS with retained SMARCB1 expression, such as SMARCA4 (BRG1), SMARCC1, or SMARCC2 [176,177].

In the prospective study from the COG and EpSSG, the 5-year event-free survival (EFS) and 5-year overall survival (OS) rates were, respectively, 61.5% and 69.2% in the COG cohort, and 79.6% and 80.6% in the EpSSG cohort [171]. The most significant prognostic factor was lymph node involvement. Other significantly unfavorable prognostic factors were a higher tumor invasiveness (T2 vs. T1), a higher IRS group (III vs. II vs. I), and a higher FNCLCC (Fédération Nationale des Centres de Lutte Contre le Cancer) grade (3 vs. 2). Proximal-type EpS is deemed to have a worse prognosis than distal Eps, however, it remains debated whether there is a real difference of aggressivity between the two, or the location in itself is responsible for the different outcomes [178,179]. In most cases, adequate complete resection is sufficient in localized tumors. Radiotherapy improves local control of locally advanced tumors. Neoadjuvant chemotherapy may be beneficial in some cases, with about 50% reported response [171].

### 7.2. Clear Cell Sarcoma of Soft Tissue

Clear cell sarcoma of soft tissue (CCS) is a very rare malignant tumor of unknown origin, occurring mainly in young adults and children. Up to 12% of cases occur before the age of 20 [180]. In children, almost half of cases occur after the age of 14, and about 85% of cases occur after the age of 10 [1]. There is no sex predominance and it seems to affect Caucasians more often than people of African or Asian descent [180]. It has a clear preference for the limbs, especially the lower limbs. The feet and ankles are typical locations [181]. CCS involves the subcutaneous tissue or deeper soft tissue. Nonetheless, it may be biopsied by dermatologists, especially when it involves the extremities. Most often, the tumor is less than 5 cm in diameter, but some cases larger than 20 cm have been reported [180]. In a review by Gonzaga et al., lymph node metastases occurred in only 6% of cases, whereas distant metastases were present in 15% of cases, most frequently in the lungs, followed by bones, liver, and brain [180]. As for most soft tissue tumors, imaging is not specific. MRI is the most useful investigation tool. Its main goal is to determine the relations of the tumor with adjacent structures [182].

On gross examination, CCS is somewhat well circumscribed but shows obvious infiltration. It is a tan-gray, firm mass [183]. Microscopically, the tumor has an infiltrative pattern of growth and is arranged in small nests or fascicles of uniform clear cells. The neoplastic cells are polygonal to fusiform with a pale eosinophilic and a round clear nucleus with a vesicular chromatin. Importantly, there is no or very slight nuclear atypia and mitoses are very rare (Figure 16A–C). CCS is characterized by the expression of melanocytic markers: S100, HMB45, and MiTF (Figure 16D). There may also be some positivity for Melan-A, NSE, and CD99. Cyclin D1 has been shown to be positive in the nucleus in about 65% of cases [184].

Similar to what is known in melanoma, there are alterations of the p16^INK4a^/p14^ARF^ pathway in CCS, which explains the decreased expression of p16 in some CCS cases, and increased expression of cyclin D1 [184]. Importantly, CCS is characterized by the t(12;22)(q13;q12) chromosomal translocation resulting in the *EWSR1-ATF1* gene fusion, in up to 90% of cases. In a small subset of cases, the t(2;22)(q34;q12) translocation resulting in the *EWSR1-CREB1* fusion gene is found. Recently, a retrospective whole-exome sequencing study of 21 CCS revealed interesting data. Deletions of *ATM* and *CHEK2* in the p53 pathway were particularly interesting. The use of poly-ADP-ribose polymerase (PARP) inhibitors (such as olaparib) or ATR inhibitors might be useful in the treatment of CCS, as reported by the Chinese team which performed in vitro treatment of CCS cell lines with such drugs [185]. However, the primary treatment of CCS is complete surgical excision with wide margins (1 cm or more for most authors). When the recommended margins cannot be ensured, adjuvant treatment is necessary, i.e., radiation therapy. Although its efficiency is debated, chemotherapy is recommended in metastatic cases, with only partial responses using anthracycline-based protocols [182].

### 7.3. Synovial Sarcoma

Synovial sarcoma is a malignant mesenchymal neoplasm of unknown origin, thus, the historical name “synovial sarcoma” is misleading. This tumor actually shows a variable epithelial differentiation and, most importantly, is characterized by a specific *SS18-SSX* fusion gene. It is mostly found in adolescents and young adults, with no gender predilection. It involves the deep soft tissues of the extremities, often in close proximity with articulations. Therefore, it falls at the limits of this review’s topic and it will not be discussed extensively. Suffice it to say that it is often a slow-growing mass that may be mistaken as benign, with no specific imaging features, although often showing areas of varied intensities on MRI, and microscopic examination may reveal either a monophasic or biphasic appearance. Monophasic synovial sarcoma is made of dense cellular sheets and fascicles of uniform spindle cells associated with a typical, but often lacking, staghorn vascular pattern. Biphasic synovial sarcoma is similar to the monophasic variant with the addition of an epithelial component, usually focal, arranged in nests or glandular structures. On immunohistochemistry, synovial sarcoma is positive for EMA and cytokeratins, but more importantly for TLE1, a useful marker, although not specific and also expressed in solitary fibrous tumors and malignant peripheral nerve sheath tumors. Most cases of synovial sarcomas also express CD99 (membranous staining) and Bcl-2. INI1 is partially or completely lost in some cases, and this should be known to avoid a misdiagnosis of rhabdoid tumor. In any case, the final diagnosis of synovial sarcoma is now made by the presence of one of the following gene fusions: *SS18-SSX1*, *SS18-SSX2*, *SS18-SSX4*, or *SS18L1-SSX1*. Synovial sarcoma has a propensity for late local recurrence and late metastasis, most often in the lungs and pleura [186]. The treatment of choice for localized disease is wide surgical excision. If not immediately possible, neoadjuvant chemotherapy for tumor reduction is recommended before surgery, most commonly with doxorubicin and/or ifosfamide. The use of local adjuvant radiation therapy is still debated [187].

## 8. Conclusions

Malignant superficial mesenchymal tumors are rare in children; however, they remain to be the most frequent group of tumors as compared with the even rarer hematologic, melanocytic, or epithelial malignancies. Recently, new entities have been described based on molecular anomalies, such as the *NTRK*-rearranged spindle cell neoplasms, the *EWSR1-SMAD3*-rearranged fibroblastic tumor, or the *ALK*-rearranged infantile fibrosarcoma-like tumor. At the same time, the complexity of genetically classifying these tumors has increased as a result of the discovery of new fusion transcripts in already well-defined entities, for example, *COL1A2-PDGFB*, *COL6A3-PDGFD*, or *EMILIN2-PDGFD* fusion-positive DFSP, the various *NTRK* fusions in infantile fibrosarcoma, or the new *FOSB* fusions in pseudomyogenic hemangioendothelioma. These advances may lead to redefinition of some tumor groups, as exemplified by the proximity between infantile fibrosarcoma and *NTRK*-rearranged spindle cell neoplasms, which may belong to a common spectrum.

While a minority of tumors remain strictly histo-clinical entities with no known genetic anomalies to date, such as PFHT, the vast majority of the tumors discussed in the present review are defined, partially in most cases, on molecular grounds. Therefore, there is a strong need for histo-molecular cooperation. This is true for diagnosis, but also more and more relevant in therapeutic approaches, as proven by the development of NTRK inhibitors in infantile fibrosarcoma, mTOR-inhibitors (sirolimus, everolimus) in KHE or in pseudomyogenic hemangioendothelioma, or even PARP inhibitors in clear cell sarcoma of soft tissues. Nevertheless, surgery remains the most valuable treatment in a lot of cases, the best example of this being, arguably, Mohs micrographic surgery in the treatment of DFSP.

Lastly, the specificities of children and skin tumors should not be forgotten: particular clinical presentations (e.g., “blueberry muffin rash” or congenital tumors); the possibility of a cancer predisposition syndrome (NF1, Li–Fraumeni, among others) or congenital immunodeficiency syndrome (e.g., Wiscott-Aldrich syndrome); and the particular behavior of some superficial tumors (e.g., the more indolent course of superficial RMS as compared with deeper RMS) should be kept in mind by all specialists dealing with skin tumors in children.

## Figures and Tables

**Figure 1 cancers-14-02160-f001:**
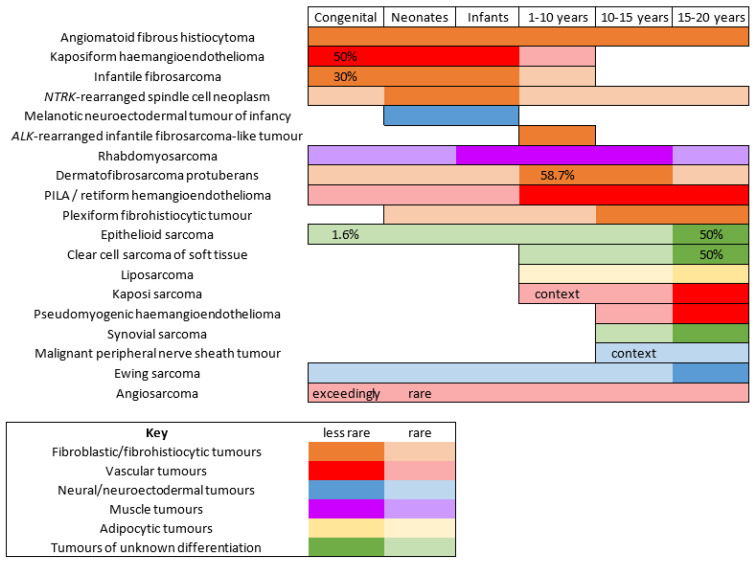
Malignant superficial mesenchymal tumors of the child by age range.

**Figure 2 cancers-14-02160-f002:**
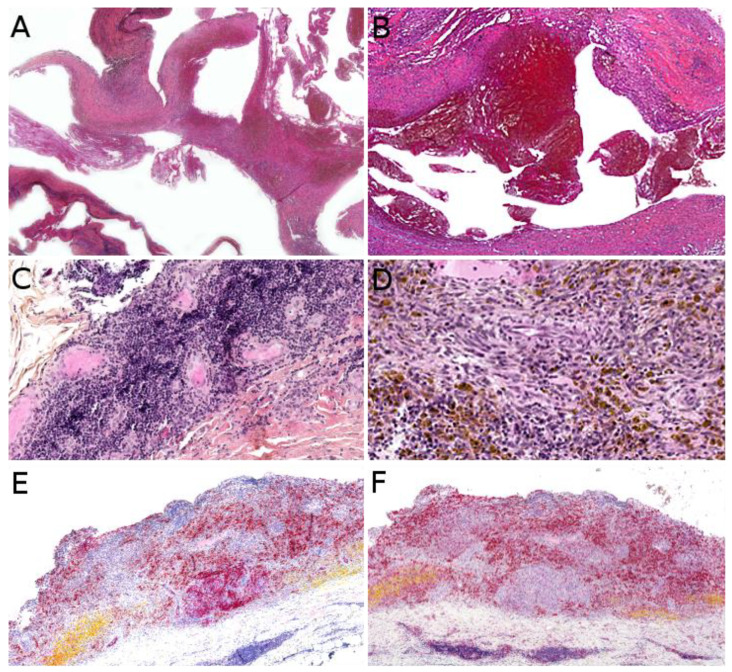
Angiomatoid fibrous histiocytoma: (**A**) Pseudo-angiomatous spaces (×25); (**B**) blood-filled pseudo-angiomatous space (×25); (**C**) dense lymphocytic infiltrate at the periphery of the tumor (×25); (**D**) histiocytoid tumor cells and hemosiderin deposits (×200); (**E**) positivity for desmin (×25) and (**F**) for CD68 (×25).

**Figure 3 cancers-14-02160-f003:**
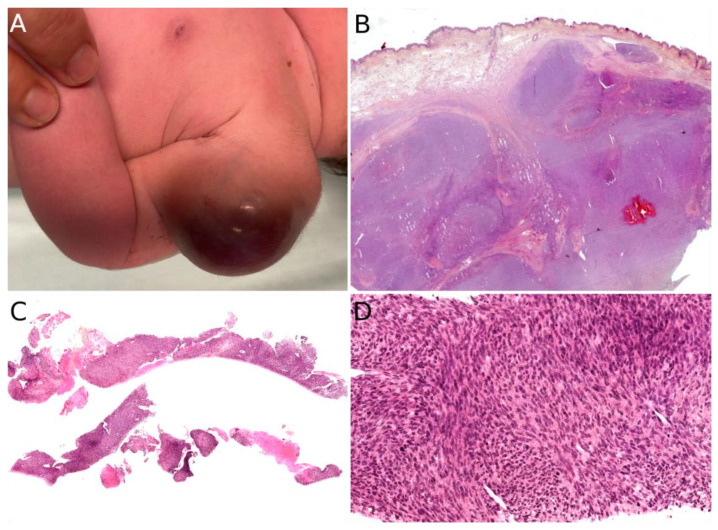
Infantile fibrosarcoma: (**A**) Large vascularized tumor on the arm of a newborn; (**B**) large surgical biopsy of a dermo-hypodermal multi-lobulated tumor with high cellularity (×25); (**C**) core-needle biopsy of another case (×25); (**D**) cellular, monomorphic spindle cell proliferation arranged in compact sheets or long fascicles (×100).

**Figure 4 cancers-14-02160-f004:**
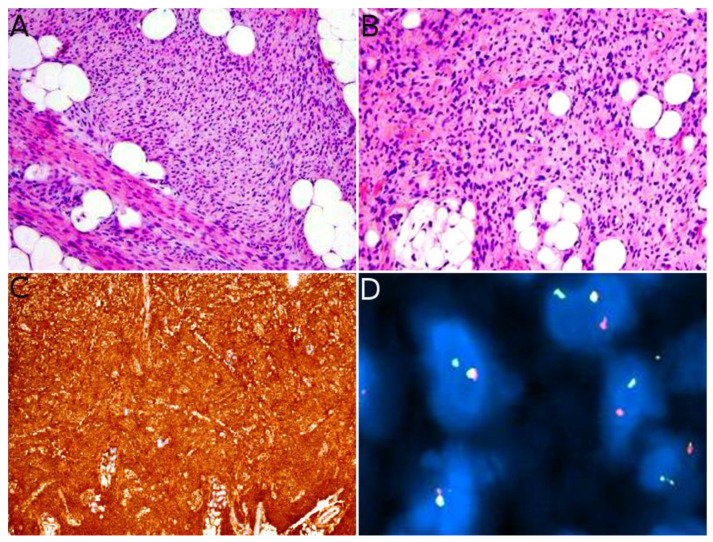
*NTRK*-rearranged spindle cell neoplasm: (**A**) Infiltrative growth pattern within the subcutaneous fat, with spindle cells haphazardly arranged or in fascicles (×100); (**B**) mild atypia and hyperchromasia (×100); (**C**) diffuse positivity for pan-TRK (×25); (**D**) fusion signals involving *NTRK* on FISH.

**Figure 5 cancers-14-02160-f005:**
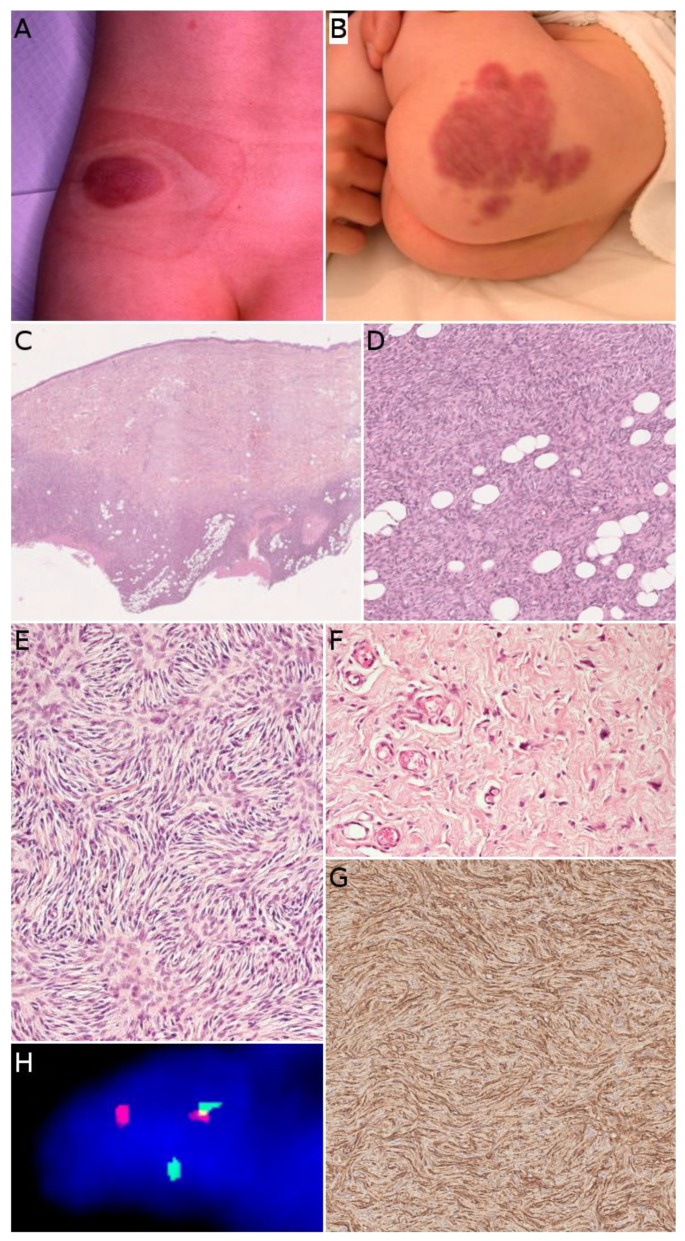
Dermatofibrosarcoma protuberans: (**A**) Plaque-like lesion on the flank of a teenager; (**B**) irregular plaque-like lesion on the buttock of neonate (congenital lesion); (**C**) dense cellular proliferation infiltrating into the subcutaneous fat (×25); (**D**) infiltration of the subcutaneous adipose tissue (×100); (**E**) spindle cells in a typical storiform pattern (×200); (**F**) giant cell fibroblastoma: less cellular tumor with giant cells and a loose stroma (×100); (**G**) diffuse strong CD34 positivity (×100); (**H**) *COL1A1-PDGFB* fusion on FISH.

**Figure 6 cancers-14-02160-f006:**
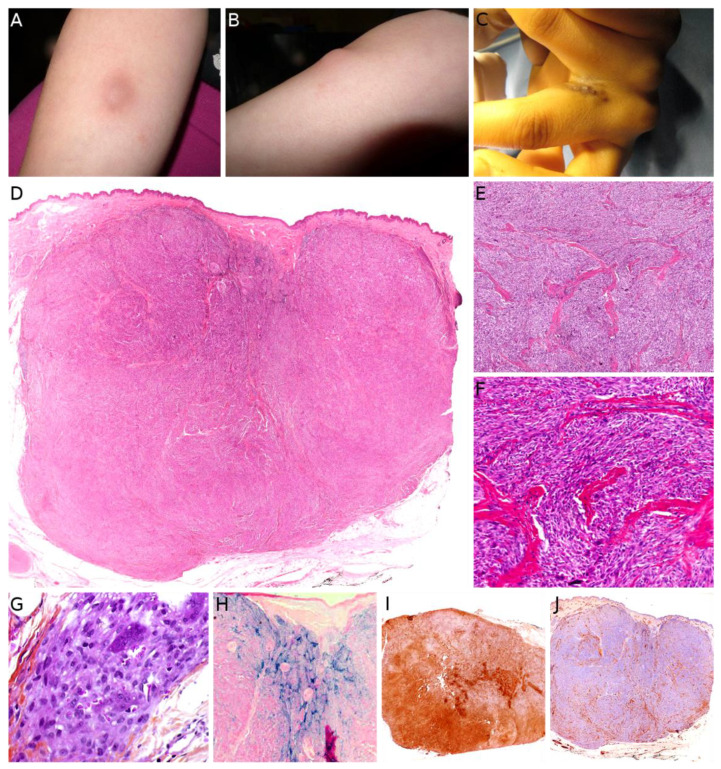
Plexiform fibrohistiocytic tumor: (**A**,**B**) Subcutaneous nodule of the leg in a girl; (**C**) irregular indurated plaque of the inter-digital space in a teenager; (**D**) dense cellular, subcutaneous, nodular proliferation extending in the dermis (×25); (**E**) plexiform pattern of growth (×100); (**F**) plexiform pattern with some multinucleated giant cells (×100); (**G**) histiocyte-like cells in a fascicle with a multinucleated giant cell (×200); (**H**) hemosiderin deposition (Perls ×25); (**I**) diffuse strong CD163 positivity (×25); (**J**) partial SMA positivity (×25).

**Figure 7 cancers-14-02160-f007:**
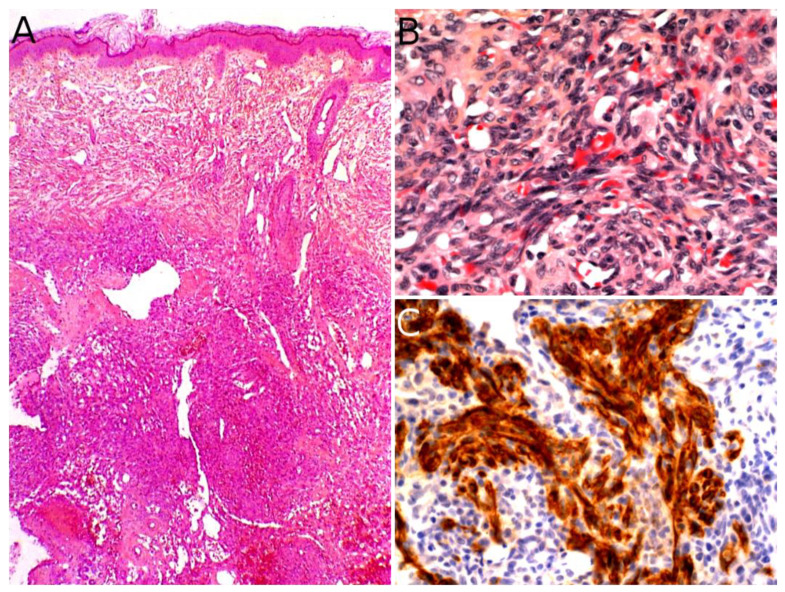
Kaposiform hemangioendothelioma: (**A**) Deep dermal and subcutaneous, infiltrating proliferation of spindle cells arranged in nodules (×25); (**B**) spindle endothelial cells with no mitoses (×200); (**C**) positivity for the lymphatic marker podoplanin in the spindle cells (×200).

**Figure 8 cancers-14-02160-f008:**
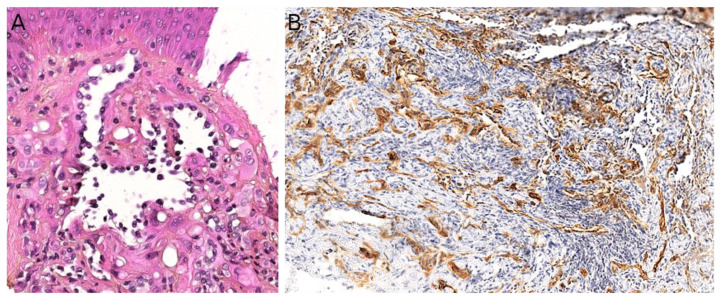
Papillary intralymphatic angioendothelioma: (**A**) Vascular proliferation with papillary tufts protruding from the vessel wall, lined by hobnail endothelial cells with large hyperchromatic nuclei (×200); (**B**) partial positivity for the lymphatic marker podoplanin (×100).

**Figure 9 cancers-14-02160-f009:**
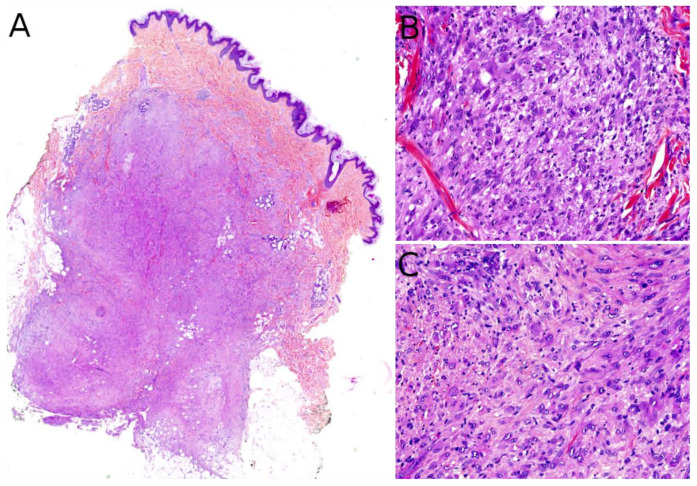
Pseudomyogenic hemangioendothelioma: (**A**) Dense cellular proliferation with an infiltrating pattern of growth (×25); (**B**) sheets of epithelioid cells or plump spindle cells with a deeply eosinophilic cytoplasm (×100); (**C**) rhabdomyoblast-like cells with moderate nuclear atypia and some neutrophils in the stroma (×200).

**Figure 10 cancers-14-02160-f010:**
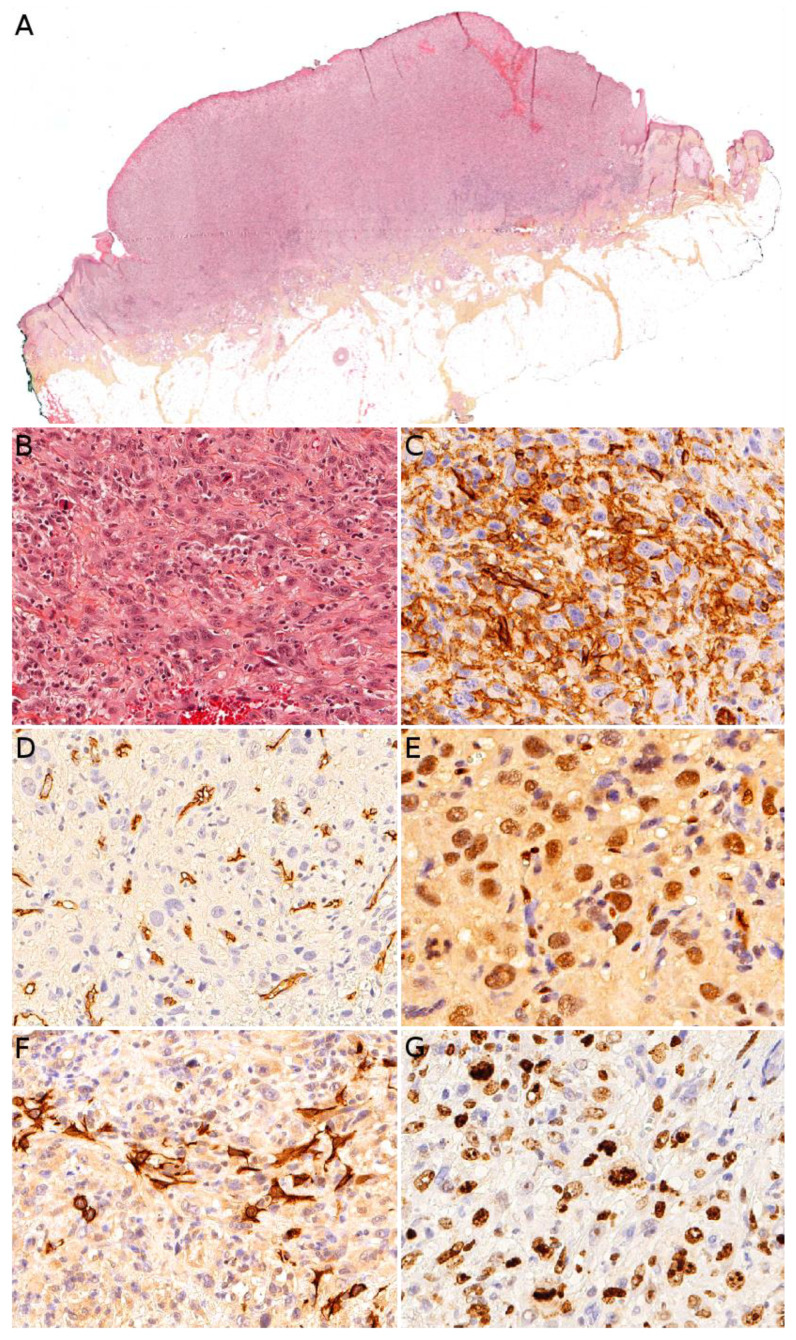
Angiosarcoma: (**A**) Ulcerated nodule (×25); (**B**) atypical epithelioid cells with numerous mitoses and a tripolar mitosis (center of the panel) (×200); (**C**) strong CD31 positivity (×200); (**D**) partial CD34 positivity on small dystrophic vascular structures (×200); (**E**) strong nuclear ERG positivity (×200); (**F**) partial positivity for the cytokeratin marker AE1/AE3 (×200); (**G**) high proliferation index and multiple mitoses (Ki67 ×200).

**Figure 11 cancers-14-02160-f011:**
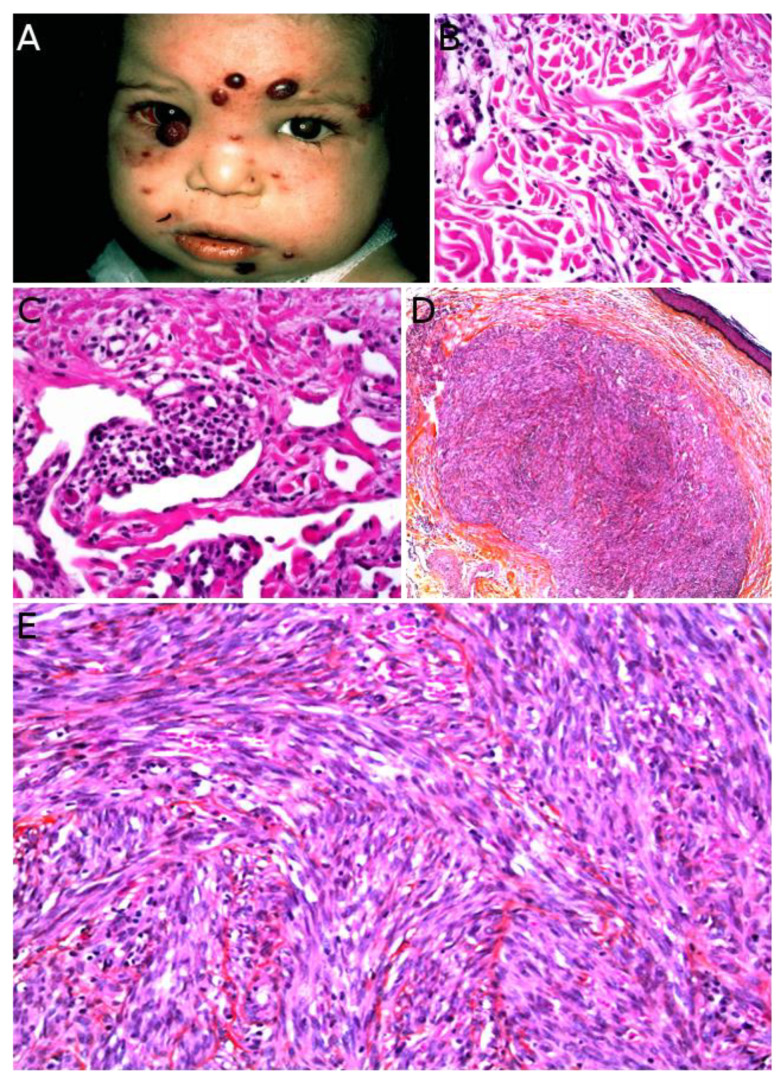
Kaposi sarcoma: (**A**) Multiple vascular nodules on the face of a child with Wiscott–Aldrich syndrome; (**B**) rare spindle cells infiltrating between the collagen bundles in a clinically macular lesion (×100); (**C**) irregular vascular spaces and lymphocytic infiltrate in a clinically papular lesion (×100); (**D**) nodular spindle-cell proliferation in a clinically nodular lesion (×25); (**E**) spindle-cell proliferation admixed with lymphocytes in a nodular stage lesion (×200).

**Figure 12 cancers-14-02160-f012:**
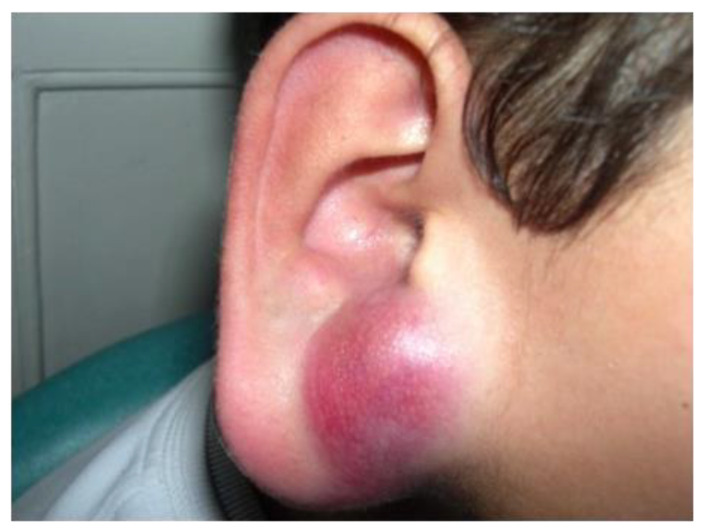
Rhabdomyosarcoma of the ear in a young boy: solitary erythematous nodule.

**Figure 13 cancers-14-02160-f013:**
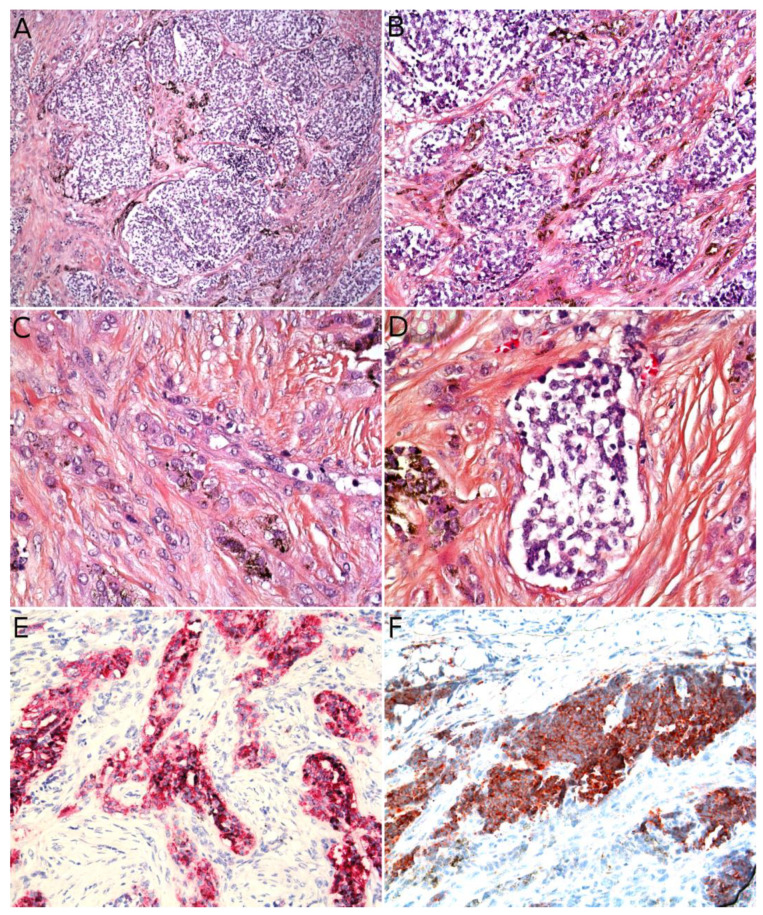
Melanotic neuroectodermal tumor of infancy: (**A**) Nests of small round blue cells surrounded by pigmented cells in a fibrous stroma (×25); (**B**) biphasic population of neuroblast-like cells and pigmented epithelioid cells (×100); (**C**) higher view of the epithelioid cells producing melanin (×200); (**D**) higher view of the neuroblast-like cells arranged in a small nest (×200); (**E**) HMB45 positivity on the epithelioid cells (×100); (**F**) synaptophysin positivity on the small neuroblast-like cells (×100).

**Figure 14 cancers-14-02160-f014:**
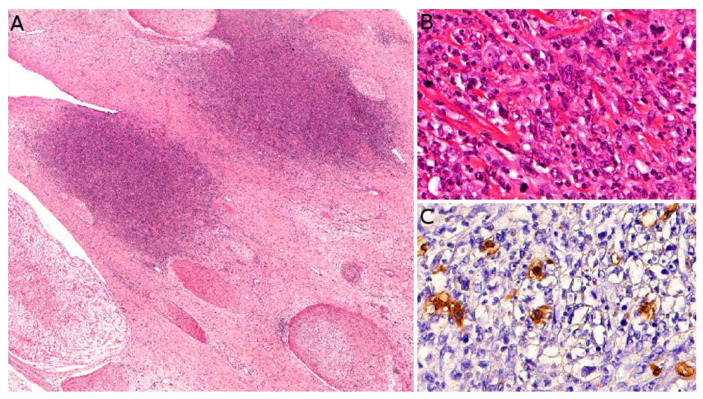
Malignant peripheral nerve sheath tumor (MPNST): (**A**) MPNST arising on a pre-existing plexiform neurofibroma in a child with NF1, note the two highly cellular areas among the otherwise poorly cellular plexiform neurofibroma (×25); (**B**) higher view of one of the MPNST areas showing nuclear atypia and numerous mitoses (×100); (**C**) partial loss of S100 expression with only sparse positive cells (×200).

**Figure 15 cancers-14-02160-f015:**
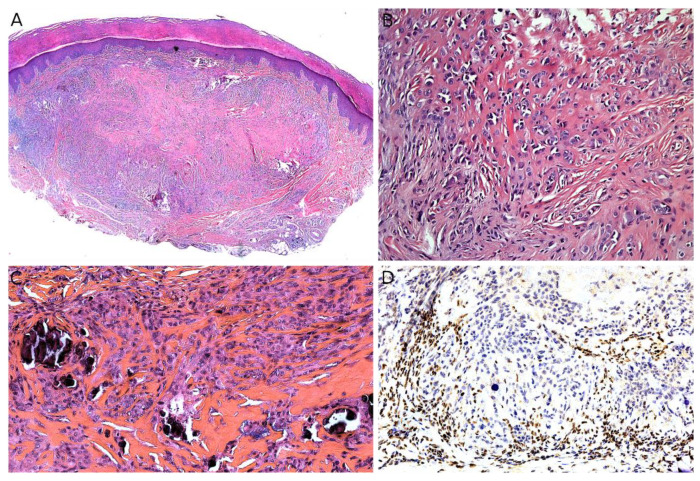
Epithelioid sarcoma: (**A**) Dermal nodule with central necrosis in an acral skin (note the thick stratum corneum) (×25); (**B**) epithelioid cells in a dense fibrous stroma (×100); (**C**) epithelioid cells and calcifications (×100); (**D**) loss of INI1 in the tumor cells (×100).

**Figure 16 cancers-14-02160-f016:**
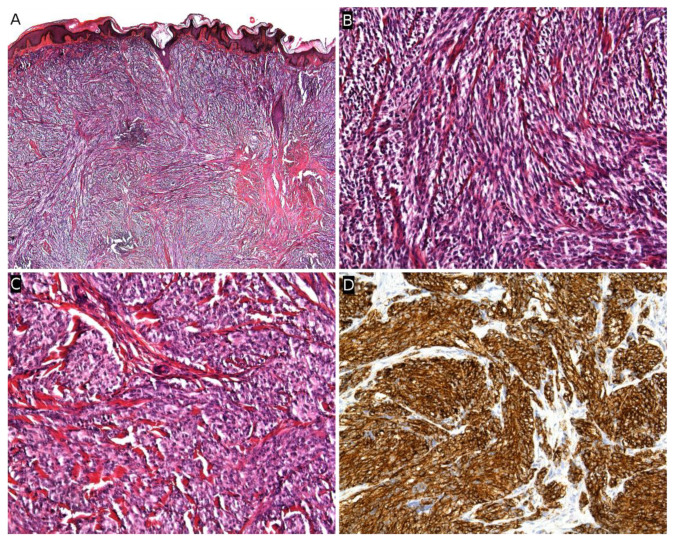
Clear cell sarcoma of soft tissue: (**A**) Densel cellular dermal and subcutaneous proliferation (×25); (**B**) uniform clear cells in fascicles (×100); (**C**) clear cells with no or slight nuclear atypia (×100); (**D**) diffuse HMB45 positivity (×100).

**Table 1 cancers-14-02160-t001:** Genetic anomalies in pediatric malignant superficial mesenchymal tumors (tumors are listed by order of appearance in the text).

Tumor	Major Genetic Anomalies	Other Genetic Anomalies
Angiomatoid fibrous histiocytoma	*EWSR1-CREB1*	*EWSR1-ATF1*, *FUS-ATF1*
Infantile fibrosarcoma	*ETV6-NTRK3*	*EML4-NTRK3*, *RBPMS-NTRK3*, *SPECC1L-NTRK3* Other gene fusions involving *NTRK1*, *NTRK2*, *MET*, *RET*, *RAF1*
*NTRK*-rearranged spindle cell neoplasms	*LMNA-NTRK1* *TPM3-NTRK1* *TPR-NTRK1*	*SQSTM1-NTRK1* *EML4-NTRK3* *KHDRBS1-NTRK3* Other gene fusions involving *NTRK1,2,3*
*EWSR1-SMAD3*-rearranged fibroblastic tumor	*EWSR1-SMAD3*	
*ALK*-rearranged infantile fibrosarcoma-like tumor	*ALK-AK5*, *ALK-ERC1*	Other gene fusions involving *ALK?*
Dermatofibrosarcoma protuberans/giant cell fibroblastoma	*COL1A1-PDGFB*	*COL1A2-PDGFB*, *COL6A3-PDGFD*, *EMILIN2-PDGFD*
Plexiform fibrohistiocytic tumor	Unknown	
Kaposiform hemangioendothelioma	-	Mutations in *GNA14*
Papillary intralymphatic angioendothelioma (PILA)/retiform hemangioendothelioma	-	
Pseudomyogenic hemangioendothelioma	*SERPINE1-FOSB*	*ACTB-FOSB*, *WWTR1-FOSB*, *CLTC-FOSB*, *EGFL7-FOSB*
Angiosarcoma	Complex anomalies (high grade tumor)	
Alveolar rhabdomyosarcoma (ARMS)	*PAX7-FOXO1*, *PAX3-FOXO1*, or no *FOXO1* fusion	
Embryonal rhabdomyosarcoma (ERMS)	Unknown	
Myxoid liposarcoma	*FUS-DDIT3*	*EWSR1-DDIT3*
Melanotic neuroectodermal tumor of infancy	unknown	*BRAF*V600E mutation (one case)
Ewing sarcoma	*EWSR1-FLI1*, *EWSR1-ERG*	*EWSR1-ETV1*, *EWSR1-ETV4*, *EWSR1-FEV*, *FUS-ERG*, *FUS-FEV*
Malignant peripheral nerve sheath tumor (MPNST)	Unknown	Loss of function mutations in *SUZ12* or *EED*
Epithelioid sarcoma	*SMARCB1* deletion, mutation or epigenetic silencing	*SMARCA4*, *SMARCC1*, *SMARCC2* deletions
Clear cell sarcoma of soft tissue	*EWSR1-ATF1*	*EWSR1-CREB1*
Synovial sarcoma	*SS18-SSX1*, *SS18-SSX2*, *SS18-SSX4*	*SS18L1-SSX1*

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
