# Peer review of "Malignant Superficial Mesenchymal Tumors in Children"

_cancers, 2022, doi:10.3390/cancers14092160_

Round 1
Reviewer 1 Report
This is a very interesting review on Malignant Superficial Mesenchymal Tumors in children.
It's well written and this work propose a very interesting and satisfying overview about this topic.
Please make a summary table about the genetic defects involved in these mesenchymal tumors, as cited in the main text
Author Response
Dear reviewer,
Thank you very much for your comments.
A table of the genetic defects has been added as requested.
Yours sincerely
Reviewer 2 Report
This article is an excellent review of malignant superficial mesenchymal tumors in children. These tumors are rare and sometimes very difficult to differentiate clinically and histopathologically. Recently, genetic analysis has become more important for differential diagnosis. This review is very useful for obtaining systematic knowledge of the epidemiology, clinical and histopathological findings, as well as information on genetic mutations and fusion genes on these tumors. I have no major modifications to suggest.
Minor comments.
1) Some abbreviations are not spelled out in the text (ALK, NTRK etc.).
2) A correspondence chart between diseases and fusion genes would make it easier to understand.
Author Response
Dear reviewer,
Thanks a lot for your comments.
A table of the genetic defects has been added, as suggested, for easier understanding.
Some abreviations of the major genes have also been clarified in the text.
Yours sincerely
Reviewer 3 Report
The authors present a broad review article on malignant mesenchymal tumors occurring in the skin of pediatric patients. The review article tackles tumors which arise definitionally primary in the dermis (DFSP) to those tumors which on rare exception may occur in children in a superficial location (e.g. leiomyosarcoma, liposarcoma). The concept of presenting a review of pediatric mesenchymal tumors with a focus on those that a dermatologist/dermatopathologist may encounter is timely given multiple recent advancements in molecular testing, new diagnoses, and targeted therapeutics. However, as outlined below, this review would be better served by revisiting its focus/scope to better target these new advances rather than the very broad approach laid out in the current form.
Major
- Needs to better pick a focus/scope within the broad topic of pediatric mesenchymal tumors of skin. Would recommended focusing more on new/evolving diagnostic entities (eg. IFS, “NTRK rearranged mesenchymal neoplasm”, EWSR1-SMAD3 fibroblastic tumor, PMHE, among others) that have updates in either diagnostic category, molecular characterization, or treatment options.
- What are your goals of the manuscript? The simple summary states “This review aims to sort out the diversity of these malignant mesenchymal tumors in children, with particular focus on clinical features that may be useful for clinicians (especially the age) and on the newest entities and genetic data.”
- To that end, could another table help highlight new entities and molecular/genetic data/treatment options?
- Would also try to focus and spent more time on tumors that are more common/relevant for the reader; this may mean deleting sections that have may occur but are exceedingly rare in the skin in children. For example, in the current for PMHE and EHE have similar space given, yet PMHE is much more relevant to this discussion and EHE is exceedingly rare in children. Indeed both cited references in EHE section do not address children directly.
- Given above comment, I would delete section 4.2 on leiomyosarcoma, 3 on EBV smooth muscle tumors, (not relevant to discussion on pediatric cutaneous mesenchymal tumors), and section 8 (while nice to have a clinical vignette it doesn’t add to this review). Also, I would only mention once in the into and only in brief the areas you cannot cover and then do not mention again. Readers will understand that the review cannot cover all mesenchymal tumors.
- It would benefit this manuscript/review to shorten/focus the breath of content and add tables/charts to aid the reader in finding the key take away points of the review.
Minor
- Use of term of primitive neuroectodermal tumor (PNET) should be omitted; please revise as this is no longer recommended terminology. Please use Ewing sarcoma. (see 5th edition WHO terminology for Ewing Sarcoma).
- Figure 2. Part E desmin is misspelled
- Line 304 – use statement such as “among others” rather than (…) to denote other possible therapies, same issue for lines 785, 851, and 1085.
- Line 492 – EpS is useful before defined.
- Lines 498-505 – paragraph on molecular finding PMHE. Ref PMID 33550637 should be added with novel EGFL7-FOSB fusion in widely metastatic PMHE in a child with multiple dermal and visceral lesions
Author Response
Dear reviewer,
Thank you very much for your constructive comments.
As you suggested, we decided to delete some sections (EHE, LMS, EBV-associated smooth muscle tumor, blueberry muffin) which were only diluting the message and were not very relevant.
On the same note, a table has been added to summarise the genetic alterations and make them more manifest. This will help the reader to get an overview of the genetic data concerning these tumors and we hope it will be sufficient to emphazise this particular point.
Concerning the minor comments :
- PNET has been replaced by Ewing sarcoma
- spelling has been checked and corrected
- suspension points have been avoided
- EpS has been defined (it was indeed a cut-paste typo)
- The reference on EGFL7-FOSB PMHE (which we missed) has been added.
Thanks again. We hope this version will meet your expectations.
Yours sincerely